# Local nutrient addition drives plant diversity losses but not biotic homogenization in global grasslands

Qingqing Chen [1,2] ✉, Shane A. Blowes [1,3], W. Stanley Harpole [1,4,5], Emma Ladouceur[1,3,6,7,8], Elizabeth T. Borer [9], Andrew MacDougall[10], Jason P. Martina [11], Jonathan D. Bakker [12], Pedro M. Tognetti [13], Eric W. Seabloom [9], Pedro Daleo [14], Sally Power [15], Christiane Roscher [1,4], Peter B. Adler [16], Ian Donohue [17], George Wheeler[18], Carly Stevens [19], G. F. Ciska Veen [20], Anita C. Risch [21], Glenda M. Wardle [22], Yann Hautier[23], Catalina Estrada [24], Erika Hersch-Green [25], Yujie Niu [26], Pablo L. Peri [27], Anu Eskelinen [1,4,28], Daniel S. Gruner [29], Harry Olde Venterink [30], Carla D'Antonio[31], Marc W. Cadotte [32], Sylvia Haider[1,33], Nico Eisenhauer [1,34], Jane Catford [35,36], Risto Virtanen [28], John W. Morgan[37], Michelle Tedder [38], Sumanta Bagchi [39], Maria C. Caldeira [40], Miguel N. Bugalho [41], Johannes M. H. Knops[42], Chris R. Dickman [43], Nicole Hagenah[44], Anke Jentsch [26], Petr Macek [45], Brooke B. Osborne[46], Lauri Laanisto [47] & Jonathan M. Chase [1,3] ✉

Nutrient enrichment typically causes local plant diversity declines. A common but untested expectation is that nutrient enrichment also reduces variation in nutrient conditions among localities and selects for a smaller pool of species, causing greater diversity declines at larger than local scales and thus biotic homogenization. Here we apply a framework that links changes in species richness across scales to changes in the numbers of spatially restricted and widespread species for a standardized nutrient addition experiment across 72 grasslands on six continents. Overall, we find proportionally similar species loss at local and larger scales, suggesting similar declines of spatially restricted and widespread species, and no biotic homogenization after 4 years and up to 14 years of treatment. These patterns of diversity changes are generally consistent across species groups. Thus, nutrient enrichment poses threats to plant diversity, including for widespread species that are often critical for ecosystem functions.

Agricultural fertilization practices and atmospheric nutrient deposition have led to increased availability and redistribution of soil nutrients globally[1–3]. At local scales (i.e., α diversity), nutrient enrichment tends to reduce plant diversity[4,5]. This diversity decline is typically ascribed to disproportionate losses of rare species (i.e., species with relatively low cover) because small populations are more susceptible to extinction[6–8]. In addition, nutrient enrichment often leads to the removal of species with traits ill-suited for effectively competing in high nutrient conditions[6]. While species can vary widely in their nutrient requirements and tolerances, groups of species with similar (shared) characteristics can be lost from a flora. For example, native species are more likely to be lost than non-native species when

nutrients are enriched because non-natives are often better-adapted to nutrient-rich conditions[6,9,10]. Similarly, nitrogen-fixing legumes may be more vulnerable than other species in high nutrient environments due to their decreased competitive advantage[6,11].

Despite clear evidence that nutrient enrichment causes losses of α diversity in grasslands[4,5], how these losses are reflected at larger spatial scales (i.e., γ diversity; calculated by aggregating local communities) is less straightforward[12–14]. Yet, it is diversity loss and change at larger spatial scales that is most often relevant for biodiversity conservation and management as well as for the provision of ecosystem functions and services[15,16]. It is often assumed that nutrient enrichment, like many other global changes, results in biotic homogenization (i.e., increasing similarity in composition among local communities, quantified as a decrease in β diversity)[17–21]. This is because local nutrient enrichment is expected to create homogeneous nutrient conditions among localities and consistently select for a smaller pool of species that are nutrient-demanding, fast-growing, and highly competitive for light[22,23]. Previous investigations of scale-dependent diversity change under nutrient enrichment have tended to be short term or limited in spatial extent[24–28]. These short-term or spatially-restricted studies have found mixed results, indicating that nutrient enrichment leads to biotic homogenization[29–31], no changes in β diversity[24,26,27,32] or even differentiation (i.e., increase in β diversity)[25,28,33–37].

A recent extension to Whittaker's multiplicative β diversity partition enables linking changes in average α diversity ($\overline{\Delta\alpha}$), γ, and β diversity (in log scale) to changes in the numbers of spatially restricted

and widespread species (Fig. 1)[12,38]. This framework illustrates how nutrient enrichment could cause biotic homogenization if local communities gain widespread species (Fig. 1: scenario I), if spatially restricted species are replaced by widespread species (Fig. 1: scenario II), or if the number of spatially restricted species decreases (Fig. 1: scenario III). Conversely, nutrient enrichment could cause biotic differentiation if local communities lose widespread species (Fig. 1: scenario IV), if widespread species are replaced by spatially restricted species (Fig. 1: scenario V), or if the number of spatially restricted species increases (Fig. 1: scenario VI). Finally, if gains or losses of species at the α and γ scale are similar (i.e., approximately equal or proportional), then we would observe no change in β diversity (1:1 diagonal line in Fig. 1). However, clear links between changes in diversity across spatial scales and changes in the number of spatially restricted and widespread species under nutrient enrichment are yet to be made.

Here, we use this framework to synthesize scale-dependent plant diversity change (for the entire community and groups of species) under nutrient enrichment using a long-term standardized experiment in 72 grasslands distributed across six continents (i.e., NutNet[39]; Fig. S1; Table S1). We use two treatments: Ambient (Control) and fertilization by nitrogen, phosphorus, and potassium together (i.e., NPK). Nutrients were added at a rate of 10 g m$^{-2}$ annually. Treatments were randomly assigned to 5 m × 5 m plots and were replicated in three or more blocks. Species cover was recorded in one 1 m × 1 m permanent subplot using a standardized protocol. At each site, α diversity is determined as the number of species in each permanent subplot (i.e., species richness), and γ diversity as the total number of species occurring in three permanent subplots (for each treatment separately). We exclude additional blocks from sites that have more than three because γ and β diversity depend on the number of local communities used. We calculate Δα as the richness difference in local communities (subplots) and Δγ as the difference in the sum of the subplots under nutrient addition relative to that of control on the log scale. That is, $\Delta\alpha = \log(\alpha_{NPK}/\alpha_{Control})$ and $\Delta\gamma = \log(\gamma_{NPK}/\gamma_{Control})$. We then calculate Δβ as Δγ minus $\overline{\Delta\alpha}$, where $\overline{\Delta\alpha}$ is the average of Δα over three blocks. Overall, we find proportionally similar species losses at local and larger scales, suggesting similar magnitudes of declines of spatially restricted and widespread species. Thus, we find no clear biotic homogenization or differentiation four years, and even up to 14 years, after nutrient additions began. Moreover, these overall patterns of little change in β diversity hold consistent across species groups.

## Results and discussion

### Changes in α-, γ-, and β diversity for the entire communities

Overall, adding nutrients decreased α and γ diversity, but it had no significant effects on β diversity (Δβ = 0.03; 95% credible interval: −0.02 to 0.08) (Fig. 2; Table S3), see also ref. 27. While we observed substantial variation in $\overline{\Delta\alpha}$, Δγ, and to a lesser extent, Δβ among sites (Table S4), we found no strong relationships between $\overline{\Delta\alpha}$, Δγ, and Δβ and distance among blocks within sites, site drought intensity, grazing intensity, productivity, or species pool that have been shown in previous literature to influence diversity change under nutrient enrichment in grasslands[24,25,34,40] (Fig. S3). On a site level, we found biotic homogenization at 24 sites, differentiation at 47 sites, and no change in β diversity at one site. However, the site-level 95% credible intervals (see Methods) overlapped 0 for all sites, suggesting no significant change in β diversity with nutrient addition (Table S4). Importantly, the overall effects of nutrient addition on α, γ, and β diversity were similar when we used effective numbers of species based on either Shannon diversity or Simpson diversity that account for species relative covers[40] (Fig. S4; Table S3). Because species richness is more strongly influenced by rare species, while Shannon and Simpson diversity increasingly weigh abundant species, this result suggests that

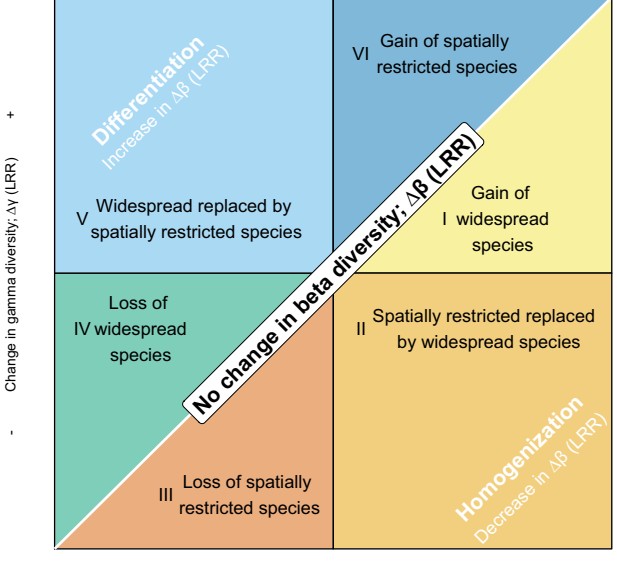

**Fig. 1 | A framework linking diversity changes at the α-, γ-, and β-scales to changes in the number of spatially restricted and widespread species.** $\overline{\Delta\alpha}$, Δγ, and Δβ are log response ratios (LRR) of average α-, γ- and β- diversity under nutrient enrichment relative to that under ambient conditions (control). Δβ is equal to zero along the 1:1 diagonal line. Below the diagonal line, Δγ < $\overline{\Delta\alpha}$, Δβ < 0, nutrient enrichment causes biotic homogenization. Above the diagonal line, Δγ > $\overline{\Delta\alpha}$, Δβ > 0, nutrient enrichment causes biotic differentiation. Moreover, homogenization can be attributed to three scenarios including I: gain of widespread species (Δγ < $\overline{\Delta\alpha}$, and Δγ > 0, $\overline{\Delta\alpha}$ > 0); II: spatially restricted species replaced by widespread species (Δγ < $\overline{\Delta\alpha}$, Δγ < 0 and $\overline{\Delta\alpha}$ > 0); III: Loss of spatially restricted species (Δγ < $\overline{\Delta\alpha}$, and Δγ < 0, $\overline{\Delta\alpha}$ < 0). Conversely, differentiation can be attributed to three scenarios that include IV: Loss of widespread species (Δγ > $\overline{\Delta\alpha}$, and Δγ < 0, $\overline{\Delta\alpha}$ < 0); V: Widespread species replaced by spatially restricted species (Δγ > $\overline{\Delta\alpha}$, Δγ > 0 and $\overline{\Delta\alpha}$ < 0); VI: gain of spatially restricted species (Δγ > $\overline{\Delta\alpha}$, and Δγ > 0, $\overline{\Delta\alpha}$ > 0). Adapted from Blowes et al.[12]. https://doi.org/10.1126/sciadv.adj9395 under a CC BY license: https://creativecommons.org/licenses/by/4.0/.

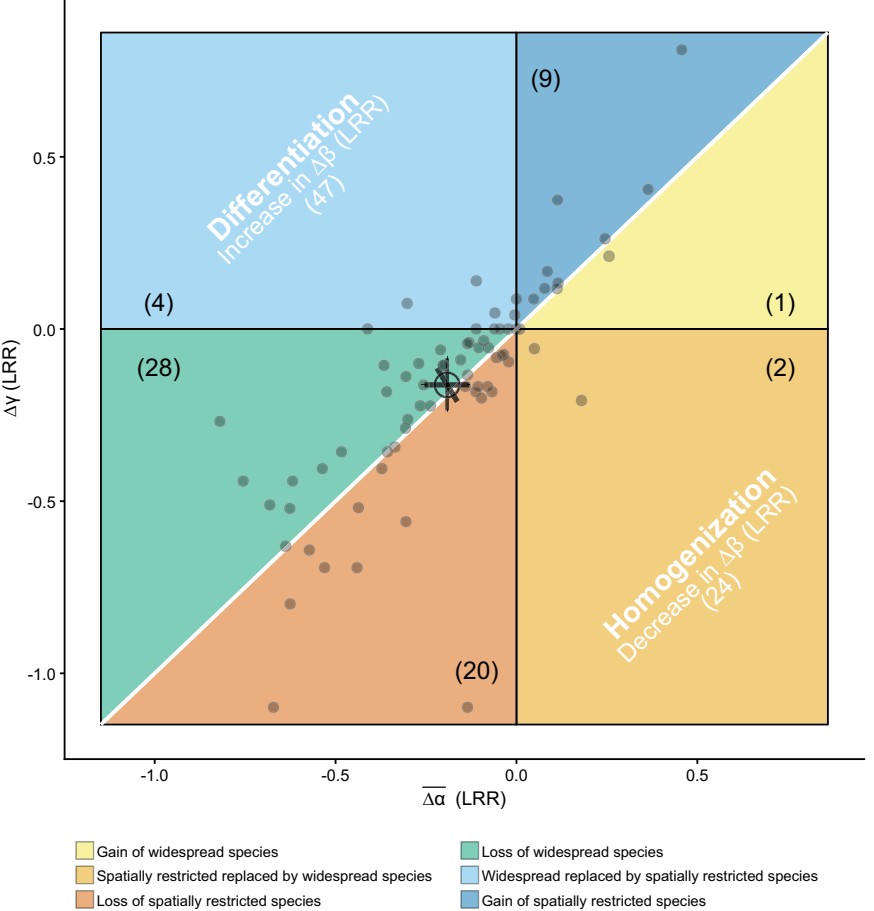

**Fig. 2 | Changes in average α, γ, and β diversity (($\overline{\Delta\alpha}$, Δγ, and Δβ) with nutrient addition.** LRR: log response ratio. The white 1:1 diagonal line indicates no effects of nutrient addition on β diversity. Numbers in the parentheses are the number of sites. When a site has $\overline{\Delta\alpha}$ = 0, Δγ = 0, or Δβ = 0, it was not counted into any of the six scenarios as shown in the framework. The small points represent site-level $\overline{\Delta\alpha}$ and Δγ at 72 sites. The large open point and error bars are the estimated mean and 95% credible intervals for $\overline{\Delta\alpha}$, Δγ, and Δβ across all sites. See Table S3 for model fit and estimated overall means and 95% credible intervals for $\overline{\Delta\alpha}$, Δγ, and Δβ. See Table S4 for site-level estimates and 95% credible intervals. Source data are provided as a Source Data file.

relatively rare and abundant species responded similarly to nutrient addition.

## Changes in α-, γ-, and β diversity for species groups

The overall proportional species loss within the community at local and larger scales on average may result if different species groups have contrasting patterns of response to nutrient addition. For instance, this result could be the case if native species loss is greater at the larger spatial scale than at the local scale, while non-native species loss is lower at the larger than the local scale. To test this possibility, we investigated changes in α, γ, and β diversity for native and non-native species separately. Extending previous studies[6,41], we found that nutrient addition decreased native species more than non-native species. Compared to non-native species, nutrient addition resulted in a 10% greater reduction of α diversity and a 16% greater reduction of γ diversity for native plant species, respectively (Table S5). The overall pattern of diversity change across spatial scales for native species largely followed that of the entire communities with similar magnitudes of decline in α and γ diversity and little change in β diversity (Δβ = 0; 95% credible interval: −0.05 to 0.06; Fig. 3A). For non-native species, overall, nutrient addition decreased α diversity 6% more than γ diversity (Table S5). But nutrient addition had no significant effects on β diversity for non-native species (Δβ = 0.04; 95% credible interval: −0.05 to 0.14; Fig. 3B).

We also separated species into forb, graminoid, legume, and woody species to investigate scale-dependent diversity change within species groups. Nutrient addition led to the greatest reduction of α

diversity for forb species and of γ diversity for woody species (Table S6). Similar to that of entire communities, nutrient addition decreased α and γ diversity by similar magnitudes and it had no effects on β diversity for graminoid species (Δβ = 0.01; 95% credible interval: −0.04 to 0.05; Fig. 4B; Table S6) and legume species (Δβ = 0.00; 95% credible interval: −0.17 to 0.18; Fig. 4C; Table S6). Overall, nutrient addition decreased α diversity 8% more than γ diversity for forb species, while it decreased γ diversity 11% more than α diversity for woody species (Table S6). However, nutrient addition also did not have significant effects on β diversity for forb species (Δβ = 0.09; 95% credible interval: −0.02 to 0.19; Fig. 4C; Table S6). Nutrient addition caused a weak biotic homogenization for woody species (Δβ = −0.14; 95% credible interval: −0.30 to 0.003; Table S6), this was primarily linked to loss of spatially restricted species (Fig. 4D).

## Robustness and limitations

We tested the robustness of our results by performing multiple sensitivity tests. We redid the analyses for the effects of nutrient addition on α, γ, and β diversity for the entire communities as well as for species groups using a subset of 14 sites that had data 14 years after nutrient additions began (Figs. S5–S7). We found that the overall effects were largely similar in these longer-term sites to that of 72 sites that had data four years after nutrient additions began. Because three spatial blocks may be limited in spatial extent for estimating effects on β diversity, we redid the analyses for the effects of nutrient addition on α, γ, and β diversity for the entire communities as well as for species groups using

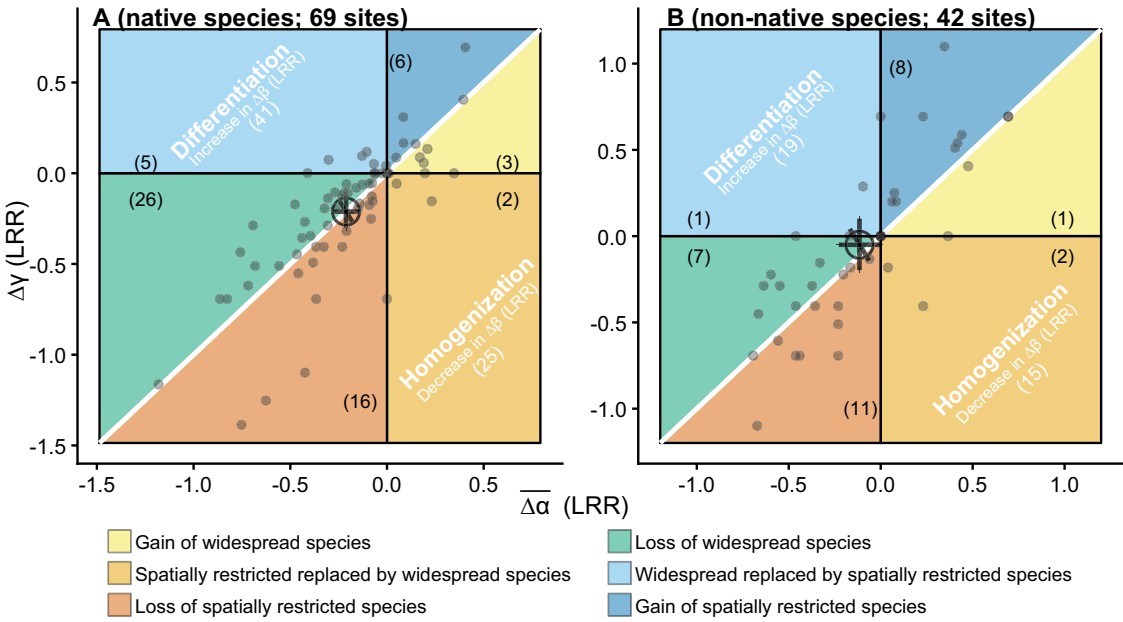

**Fig. 3 | Changes in average α, γ, and β diversity ($\overline{\Delta\alpha}$, Δγ, and Δβ) with nutrient addition for native and non-native species groups. A** Native and **B** non-native species. LRR: log response ratio. The white 1:1 diagonal line indicates no effects of nutrient addition on β diversity. Numbers in the parentheses are the number of sites. When a site has $\overline{\Delta\alpha}$ = 0, Δγ = 0, or Δβ = 0, it was not counted into any of the six scenarios as shown in the framework. The small points represent site-level $\overline{\Delta\alpha}$ and Δγ. The large open point and error bars are the estimated mean and 95% credible intervals for $\overline{\Delta\alpha}$, Δγ, and Δβ across all sites. See Table S5 for model fit and estimated overall means and 95% credible intervals for $\overline{\Delta\alpha}$, Δγ, and Δβ. Source data are provided as a Source Data file.

11 sites that had five spatial blocks (Figs. S8–S10). Again, the overall effects were largely similar to that we found for the full analysis of the 72 sites with three blocks.

Despite our evidence for little change in β diversity with nutrient addition across the entire experiment, we recognize limitations of extrapolating these results to the landscape scale (e.g., >1 km × 1 km). This is because many ecological processes that directly and indirectly influence plant diversity can be very different at the landscape scales[14,42]. The scale at which we inferred changes in the number of spatially restricted and widespread species, by examining how many local communities they were lost from, is a relatively small spatial scale. Linking estimates of species' geographic range size and other key traits with changes in plant diversity across larger spatial scales[19,42] will deepen our understanding of the mechanisms of diversity change.

To summarize, we provide a comprehensive synthesis of the impact of nutrient addition on scale-dependent plant diversity change in grasslands by applying a framework to a globally distributed long-term experiment. The framework links changes in species richness across scales to that changes in the numbers of spatially restricted and widespread species. Overall, we found similar proportional plant diversity declines at local and larger spatial scales with nutrient addition, and little evidence for either biotic homogenization or differentiation within sites. These overall patterns were largely consistent for diversity metrics that incorporate relative species covers, across species groups, and over long time periods. This demonstrates that nutrient enrichment poses a potential threat to all plant species groups, including widespread and native species that often drive ecosystem functions and services.

## Methods
### Experimental setup
The experimental sites used in this study are part of the Nutrient Network (NutNet, Fig. S1 and Table S1). The experimental design includes a factorial manipulation of nutrients (N, P, and K) plus two fences to exclude herbivores, see ref. 39 for more details. For the analyses here, we used plots under two treatments: Ambient (Control)

and fertilization by nitrogen, phosphate, and potassium together (i.e., NPK). Treatments were randomly assigned to 5 m × 5 m plots and were replicated in three or more blocks. A micronutrient mix consists of Fe (15%), S (14%), Mg (1.5%), Mn (2.5%), Cu (1%), Zn (1%), B (0.2%), and Mo (0.05%) was added once only at the start of the experiment (i.e., year 1) for the nutrient addition plots, but not in subsequent years to avoid toxicity. Nitrogen, phosphate, potassium were added annually before the growing season of each treatment year at most sites. Nitrogen was added as $10\,\mathrm{g\,m^{-2}\,yr^{-1}}$ time-release urea [$(NH_2)_2CO$], phosphate was added as $10\,\mathrm{g\,m^{-2}\,yr^{-1}}$ triple-super phosphate [$Ca(H_2PO_4)_2$], while potassium was added as $10\,\mathrm{g\,m^{-2}\,yr^{-1}}$ sulfate [$K_2SO_4$].

Data were retrieved from the NutNet database in November 2023. We analyzed data from 72 sites where 1) nutrients were applied for at least four years; and 2) each site had at least three blocks. These sites are distributed across six continents and include a wide range of grassland types. See Fig. S1 and Table S1 for details of geolocation, grassland types, and experimental years used.

### Sampling protocol
Scientists at NutNet sites followed standard sampling protocols[39]. Specifically, a 1 m × 1 m subplot within each plot was permanently marked for annual recording of plant species composition. Species cover (%) was estimated visually for individual species in the subplots; thus the total cover of living plants may sometimes exceed 100% for multilayer canopies. At most sites, cover was recorded once per year at peak biomass. At some sites with strong seasonality, cover was recorded twice per year to include a complete list of species. For those sites, the maximum cover for each species and total biomass were used in the analyses. When taxa could not be identified to the species level, they were aggregated at the genus level but referred to as "species" for simplicity.

### Quantifying changes in α, γ, and β diversity
We measured α and γ diversity using species richness (i.e., number of species) because it is the most commonly examined diversity metric[43]. At each site, α diversity was estimated as the number of species in each permanent subplot (1 m × 1 m), and γ diversity as the total number of

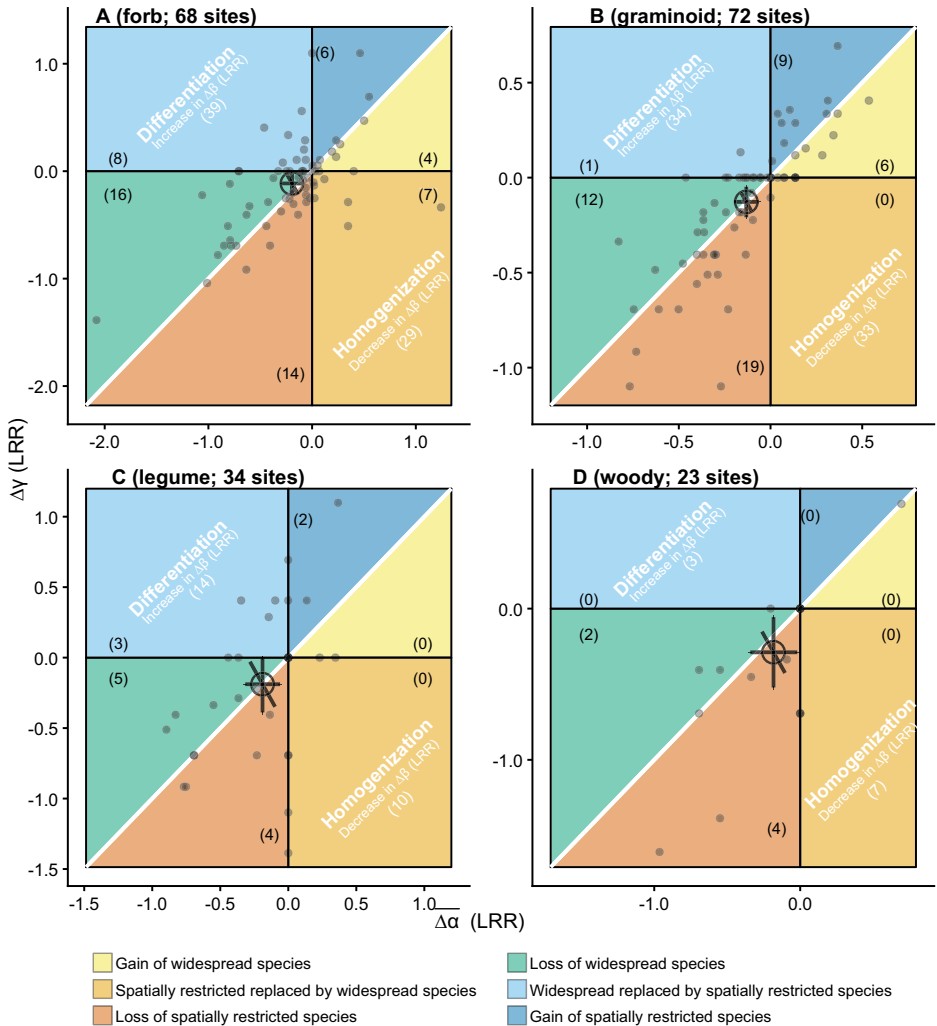

**Fig. 4 | Changes in average α, γ, and β diversity ($\overline{\Delta\alpha}$, Δγ, and Δβ) with nutrient addition for different functional species groups. A** Forb, **B** graminoid, **C** legume, and **D** woody species. LRR: log response ratio. The white 1:1 diagonal line indicates no effects of nutrient addition on β diversity. Numbers in the parentheses are the number of sites. When a site has $\overline{\Delta\alpha} = 0$, Δγ = 0, or Δβ = 0, it was not counted into any of the six scenarios as shown in the framework. The small points represent site-level $\overline{\Delta\alpha}$ and Δγ. The large open point and error bars are the estimated mean and 95% credible intervals for $\overline{\Delta\alpha}$, Δγ, and Δβ across all sites. See Table S6 for model fit and estimated overall means and 95% credible intervals for $\overline{\Delta\alpha}$, Δγ, and Δβ. Source data are provided as a Source Data file.

species occurring in three permanent subplots (for each treatment separately). To standardize sampling effort, for sites with more than three blocks, we selected the first three blocks according to the block number recorded by site PIs. The framework relies on Whittaker's multiplicative β diversity partition, and it quantifies β diversity using the effective number of communities[12]. As such, if all subplots share the same species, then β diversity would equal to one. In contrast, if each subplot has unique species, then β diversity would equal to three. We calculated Δα as the richness difference in local communities (subplots) and Δγ as the difference in the sum of the subplots under nutrient addition relative to that of control treatment on the log scale. That is, $\Delta\alpha = \log(\alpha_{NPK}/\alpha_{Control})$ and $\Delta\gamma = \log(\gamma_{NPK}/\gamma_{Control})$. We calculated Δβ as Δγ minus $\overline{\Delta\alpha}$, where $\overline{\Delta\alpha}$ is the average of Δα over three blocks. A decrease in Δβ indicates nutrient addition causes species composition to be more similar among three subplots than that among control subplots. Because sites are not evenly distributed around the world, many sites are aggregated in North America, we checked spatial autocorrelation of diversity change under nutrient addition using Moran's I[44]. We found that $\overline{\Delta\alpha}$, Δγ, and Δβ did not appear to be more similar for sites that are closer to each other (Table S2).

We fitted multilevel (also referred as mixed effects or hierarchical) models for Δα, Δγ, and Δβ (as the response variable; all on the log

scale) separately. We included random intercept for each site, model was coded as: richness change ~ 1 + (1 |sites) to estimate site-level variation. We used Bayesian analysis because it yields full posterior distributions of parameters rather than point estimates and p-values, which provides a deeper understanding of the uncertainty and variability in the results[45]. Models described above were fitted using the Hamiltonian Monte Carlo (HMC) sampler in Stan and coded using the package 'brms' (version 2.21.0) in R (version 4.4.1)[46,47]. Models were fitted without explicitly specifying priors, allowing brms to assign its default priors. Models were fitted with 6 chains and 3000 iterations (1000 iterations for warm up). Visual inspection of the HMC chains and Rhat summaries showed model convergence (all Rhats <1.03; Tables S3, S5 and S6). We visually checked posterior predictive plots to determine how well models can reproduce the data (Fig. S2).

To examine whether diversity changes were sensitive to species relative covers, we redid the above analyses (i.e., based on species richness) using Shannon diversity and Simpson diversity (both converted to effective numbers)[48] (Fig. S4). Species richness is most sensitive to rare species, followed by Shannon diversity, and Simpson diversity is more sensitive to the numbers of relatively abundant species. We calculated the exponential of Shannon diversity and the inverse form of Simpson diversity using the R package vegan (version

2.6-6.1)[49]. These three diversity metrics equal to diversity with order q = {0, 1, 2}, where increasing q decreases the influence of rare species, and $D_q = \left(\sum_{i=1}^{s} p_i^q\right)^{1/(1-q)}$, where p is the relative cover of species i, s is the total number of species. These diversity metrics are also referred to as Hill numbers[48,50].

**Site covariates.** We investigated whether the effects of nutrient addition on $\overline{\Delta\alpha}$, γ, and β diversity based on species richness were mediated by site characteristics. We included site characteristics that have been shown in previous literature to influence Δα, Δγ, and Δβ in grasslands: site species pool, site productivity, drought intensity, and grazing intensity[24,25,34,40]. We quantified drought intensity as the sum of annual evapotranspiration/precipitation, and averaged it from year 0 to 4 at each site. Precipitation and potential evapotranspiration used were downloaded from https://crudata.uea.ac.uk/cru/data/hrg/cru_ts_4.07/. We quantified the site species pool as the total number of species and site productivity as the average aboveground biomass from year 0 to 4 under the control treatment in the three blocks at each site. Aboveground biomass was harvested within two $1 \times 0.1$ m strips (in total 0.2 m$^2$), strips were moved from year to year to avoid resampling the same location. For subshrubs and shrubs occurring within strips, we collected all leaves and current year's woody growth. All biomass was dried at 60 °C (to constant mass) before weighing to the nearest 0.01 g. We used published methods to quantify an integrated grazing intensity metric from vertebrate herbivores at each site. Specifically, herbivore species (>2 kg) that consume grassland biomass were documented at each site by site PIs, and each species was assigned an importance value from 1 (present but low impact and frequency) to 5 (high impact and frequency). An index value was calculated for each site as the sum of herbivore importance values for all herbivores following refs. 51,52. We also investigated relationships between change in diversity and distance among blocks, because species composition may become less similar as the distance between sampled communities increases. The average pairwise distance among the three blocks within sites ranged from 23.04 to 12538.09 m, with a mean of 513.01 m and a median of 118.7 m across 54 sites that have geolocation data for each block. We first calculated three Euclidean distances between pairs of blocks, we then used the mean of these pairwise distances as the average distance among blocks. We used the average distance among blocks instead of area, because blocks are arranged in parallel at some sites. We fitted linear regression models with $\overline{\Delta\alpha}$, Δγ, and Δβ as the response variable separately, and each of the site characteristics was used as a predictor variable.

**Species groups.** We then investigated the effects of nutrient addition on α, γ, and β diversity within groups of species with similar characteristics following the method for changes in α, γ, and β diversity in the entire communities. We eliminated sites where no species occurred in control, nutrient addition, or both plots for a particular group because the value of the log (0) is undefined. We ran the analyses separately for native and non-native species. Native and non-native species were classified by site PIs. Then, we investigated effects of nutrient addition on species richness for different life forms including forb, graminoid, legume, and woody species because previous studies have shown that different life forms may show distinct responses to nutrient addition[6,11,53].

**Sensitivity test.** We tested whether effects of nutrient addition on species richness across spatial scales depend on experimental duration because a few single-site experiments have shown that the effects of nutrient additions on changes in diversity, especially β diversity, may take several years to emerge[29,31]. To that end, we used a subset of 14 sites that had data 14 years after nutrient additions began. Also, because three blocks may be limited in spatial extent, we tested whether combining more blocks to create the γ scale would alter our

results. We redid the analyses using data from 11 sites that had five spatial blocks.

**Reporting summary**
Further information on research design is available in the Nature Portfolio Reporting Summary linked to this article.

## Data availability
The species cover and species richness data, site abiotic and biotic environmental data used and generated in this study have been deposited in the Figshare database and are publicly available (https://doi.org/10.6084/m9.figshare.26412295.v4). The NutNet data are publicly available on the Environmental Data Initiative (EDI) (https://portal.edirepository.org/nis/advancedSearch.jsp). Source data are provided with this paper.

## Code availability
The R codes used to produce results in this study have been deposited in the GitHub (https://github.com/chqq365/plant-diversity-and-biotic-homogenization.git) and archived through Zenodo (https://doi.org/10.5281/zenodo.14902812).

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

## Acknowledgements

This work was generated using data from the Nutrient Network (http://www.nutnet.org) experiment, funded at the site-scale by individual researchers. We thank the Minnesota Supercomputer Institute for hosting project data and the Institute on the Environment for hosting Network meetings. We thank researchers from the NutNet who contributed data to our analysis, Table S7 lists these researchers. We thank Shuli Niu for constructive suggestions to improve our manuscript. Nitrogen fertilizer was donated to NutNet by Crop Production Services, Loveland, CO. Funding: German Center for Integrative diversity Research (iDiv) Halle-Jena-Leipzig (DFG FZT 118-202548816) for the contribution of J.M.C., S.A.B., E.L., Q.C., W.S.H., and other iDiv co-authors. ERC Advanced Grant (MetaChange) to J.M.C. for the contribution of J.M.C., S.A.B. Views and opinions expressed are however those of the author(s) only and do not necessarily reflect those of the European Union or the European Research Council. Neither the European Union nor the granting authority can be held responsible for them. National Science Foundation grant NSF-DEB-1042132 (E.T.B., E.W.S.; for NutNet coordination and data management) National Science Foundation grant NSF-DEB-1234162 (E.T.B., E.W.S.; for Long-Term Ecological Research at Cedar Creek). National Science Foundation grant NSFDEB-1831944 (E.T.B., E.W.S.; for Long-Term Ecological Research at Cedar Creek), and the Institute on the Environment (DG-0001-13; E.T.B., E.W.S.).

## Author contributions

Q.C., J.M.C., S.A.B., E.L., and W.S.H. conceived the study; Q.C., J.M.C., and S.A.B. developed the methodology; Q.C. analyzed data with contribution from S.A.B.; W.S.H., E.T.B., A.M., J.P.M., J.D.B., P.M.T., E.W.S., P.D., S.A.P., C.R., P.B.A., I.D., G.W., C.S., G.F.V., A.C.R., G.M.W., Y.H., C.E., Y.N., P.L.P., A.E., D.S.G., H.O., C.D., M.W.C., S.H., N.E., J.C., R.V., J.W.M., M.T., S.B., M.C.C., M.B., J.M.H.K., C.R.D., N.H., A.J., P.M., B.B.O., and L.L. contributed data; Q.C. visualized the results; Q.C. wrote the original draft of the manuscript; Q.C., S.A.B., W.S.H., E.L., E.T.B., A.M., J.P.M., J.D.B.,

P.M.T., E.W.S., P.D., S.A.P., C.R., P.B.A., I.D., G.W., C.S., G.F.V., A.C.R., G.M.W., Y.H., C.E., E.H., Y.N., P.L.P., A.E., D.S.G., H.O., C.D., M.W.C., S.H., N.E., J.C., R.V., J.W.M., M.T., S.B., M.C.C., M.B., J.M.H.K., C.R.D., N.H., A.J., P.M., B.B.O., L.L., and J.M.C. reviewed and edited the manuscript. See Table S8 for more details for the contribution of each author.

## Funding

## Competing interests
The authors declare no competing interests.

## Additional information

[1]German Centre for Integrative Biodiversity Research (iDiv) Halle-Jena-Leipzig, Puschstraße 4, 04103 Leipzig, Germany. [2]Senckenberg Museum for Natural History Görlitz, D-02826 Görlitz, Germany. [3]Department of Computer Science, Martin Luther University Halle-Wittenberg, Halle (Saale), Germany. [4]Helmholtz Center for Environmental Research—UFZ, Department of Physiological Diversity, Permoserstrasse 15, 04318 Leipzig, Germany. [5]Martin Luther University Halle-Wittenberg, Am Kirchtor 1, 06108 Halle (Saale), Germany. [6]Department of Biology, University of Prince Edward Island, Charlottetown, PE, Canada. [7]Canadian Centre for Climate Change and Adaptation, University of Prince Edward Island, St. Peter's Bay, Charlottetown, PE C0A 2A0, Canada. [8]School of Climate Change and Adaptation, University of Prince Edward Island, Charlottetown, PE C1A 4P3, Canada. [9]Department of Ecology, Evolution, and Behavior, University of Minnesota, St. Paul, MN 55108, USA. [10]Department of Integrative Biology, University of Guelph, Guelph, ON N1G2W1, Canada. [11]Department of Biology, Texas State University, San Marcos, TX 78666, USA. [12]School of Environmental and Forest Sciences, University of Washington, Seattle, WA, USA. [13]Instituto de Investigaciones Fisiológicas y Ecológicas Vinculadas a La Agricultura (IFEVA), Facultad de Agronomía, Universidad de Buenos Aires and CONICET, Buenos Aires, Argentina. [14]Instituto de Investigaciones Marinas y Costeras (IIMyC), UNMdP-CONICET, CC 1260 Correo Central, B7600WAG Mar Del Plata, Argentina. [15]Hawkesbury Institute for the Environment, Locked Bag 1797, Penrith, NSW 2751, Australia. [16]Department of Wildland Resources and the Ecology Center, Utah State University, Logan, UT 84322, USA. [17]Zoology, School of Natural Sciences, Trinity College Dublin, Dublin, Ireland. [18]University of Nebraska-Lincoln, Lincoln, NE, USA. [19]Lancaster Environment Centre, Lancaster University, Lancaster LA1 4YQ, UK. [20]Department of Terrestrial Ecology, Netherlands Institute of Ecology, Droevedaalsesteeg 10, 6708 PB Wageningen, The Netherlands. [21]Swiss Federal Institute for Forest, Snow and Landscape Research WSL, Zuercherstrasse 111, 8903 Birmensdorf, Switzerland. [22]Desert Ecology Research Group, School of Life and Environmental Sciences, ARC Training Centre in Data Analytics for Resources and Environments (DARE), The University of Sydney, Sydney, NSW 2006, Australia. [23]Ecology and Biodiversity Group, Department of Biology, Utrecht University, Padualaan 8, 3584 CH Utrecht, The Netherlands. [24]Department of Life Sciences, Imperial College London, Silwood Park Campus, Buckhurst Road, Ascot SL5 7PY, UK. [25]Department of Biological Sciences, Michigan Technological University, Houghton, MI 49930, USA. [26]Disturbance Ecology and Vegetation Dynamics, Bayreuth Center of Ecology and Environmental Research (BayCEER), University of Bayreuth, Bayreuth, Germany. [27]Instituto Nacional de Tecnología Agropecuaria (INTA), Universidad Nacional de La Patagonia Austral (UNPA), CONICET, Río Gallegos, Santa Cruz, Argentina. [28]Ecology & Genetics Unit, University of Oulu, Oulu, Finland. [29]Department of Entomology, University of Maryland, College Park, MD, USA. [30]Vrije Universiteit Brussel, Department Biology, WILD, Pleinlaan 2, 1050 Brussels, Belgium. [31]University of California, Santa Barbara, Santa Barbara, CA 93106, USA. [32]Department of Biological Sciences, University of Toronto Scarborough, Toronto, ON, Canada. [33]Institute of Ecology, Leuphana University of Lüneburg, Universitätsallee 1, 21335 Lüneburg, Germany. [34]Institute of Biology, Leipzig University, Puschstrasse 4, 04103 Leipzig, Germany. [35]Department of Geography, King's College London, 30 Aldwych, London WC2B 4BG, UK. [36]School of Ecosystem & Forest Sciences, University of Melbourne, Parkville, VIC 3010, Australia. [37]Department of Environment & Genetics, La Trobe University, Bundoora, VIC 3083, Australia. [38]Centre for Functional Biodiversity, School of Life Sciences, University of KwaZulu-Natal, Pietermaritzburg, South Africa. [39]Centre for Ecological Sciences, Indian Institute of Science, Bangalore, India. [40]Forest Research Centre, Associate Laboratory TERRA, School of Agriculture, University of Lisbon, Lisbon, Portugal. [41]Centre for Applied Ecology 'Prof. Baeta Neves' (CEABN-InBIO), School of Agriculture, University of Lisbon, Lisbon, Portugal. [42]Health & Environmental Sciences Department, Xi'an Jiaotong-Liverpool University, Suzhou, Jiangsu, China. [43]Desert Ecology Research Group, School of Life and Environmental Sciences, The University of Sydney, Sydney, NSW 2006, Australia. [44]Mammal Research Institute, Department of Zoology and Entomology, University of Pretoria, Pretoria, South Africa. [45]Institute of Hydrobiology, Biology Centre of the Czech Academy of Sciences, Na Sadkach 7, Ceske Budejovice 370 05, Budejovice, Czech Republic. [46]Department of Environment and Society, Utah State University, Moab, UT, USA. [47]Chair of Biodiversity and Nature Tourism, Estonian University of Life Sciences, Kreutzwaldi 5, 51006 Tartu, Estonia. ✉e-mail: chqq365@gmail.com; jonathan.chase@idiv.de

