## [Peer Review file · Nature Communications]

Local nutrient addition drives plant biodiversity losses but not biotic homogenization in global grasslands

Corresponding Author: Dr Qingqing Chen

Version 0:

Reviewer comments:

Reviewer #1

(Remarks to the Author)

The experimental study of diversity loss in plant communities in response to fertilization is not new. The focus on how diversity may change in response to fertilization at multiple spatial scales, alter beta-diversity, and potentially affect biotic homogenization is more recent but is addressed by a fair amount of previous work. The more novel contribution that this paper makes is the application to the Nutrient Network Data of a new conceptual framework/topology to explore how different kinds of scale-dependent diversity changes may affect biotic homogenization/differentiation (recently published in Science Advances paper; Blowes et al. 2024).

The framework/topology of Blowes et al. (2024) provides a powerful tool for exploring how plant diversity losses, observed in fertilization studies such as NutNet, may vary in magnitude with spatial scale (alpha vs. gamma). The framework provides a robust means to test the hypothesis that enrichment will lead to compositional homogenization through the loss of localized species that exceed the loss of widespread species.

In this study, evaluation of relative changes in alpha and gamma diversity in response to experimental nutrient enrichment (fertilized versus control plots) showed no evidence for biotic homogenization or differentiation, indicating that the magnitude of species losses to fertilization for localized and widespread species were similar across the 72 sites. This is an important result and runs at odds to common expectation for plant communities.

Is this study original? Yes and no. Yes because it is the first study I know of that provides a robust general test of the homogenization hypothesis for plant communities exposed to nutrient enrichment. No because the Blowes et al paper (2024), which has some of the same authors, develop the framework and applied it in a somewhat similar manner to test for homogenization/differentiation using temporal change data from 461 metacommunities. On balance though I think this manuscript can be an important paper with wide potential interest and impact because of its focus on advancing understanding of mechanisms underlying scale-dependent plant diversity change under eutrophication.

Overall, the manuscript was well-written and clear in presentation. I could not find typos or significant grammatical problems. The basic analytical approach/philosophy is sound, and the statistical modeling seems robust, although some aspects are a bit beyond my knowledge to judge. I appreciate the effort to explore the potential site differences with covariates, analyses to explore potential differences among functional groups, and the sensitivity analyses. I see no flaws in interpretation or conclusions. Overall, a good paper worthy of publication.

(Remarks on code availability)

Reviewer #2

(Remarks to the Author)

This paper quantifies how nutrient addition affects beta diversity, including discussion of the implications of their findings for biotic homogenization. The authors build on previous work showing that both rare and common species are extirpated due to nutrient addition. Namely, the authors show a similar magnitude of reduction in gamma vs. alpha diversity declines,

suggesting that beta diversity does not change (and thus biotic homogenization is not occurring). I have a number of concerns, as well as some minor issues.

Major issues:

1. The authors could improve the connection between beta diversity and biotic homogenization. The paper's introduction is framed around the idea of biotic homogenization, but Figure 1 and the vast majority of the analyses are based on beta diversity, meaning that the connection with biotic homogenization is at times unclear.
2. Further, the focus in the paper is on losses of range-restricted species (where they assume that locally sparse species are range-restricted), but most treatments of biotic homogenization show that invasive species are a key driver. I see that the authors try to say early on that invasive species are not the key driver in this case (in lines 84-86), but this connection in those sentences is not at all explicit. Further, because biotic homogenization includes changes in both invasive and native plants, it's not at all clear to me how the analyses described in the paragraph beginning on line 157 connect with biotic homogenization. I think there needs to be some clarification of hypotheses here, much earlier in the paragraph. (e.g., "If invasive species are driving biotic homogenization, beta diversity should..."). (note that I would suggest the same for the treatment of life forms, where again the hypotheses are not clear until almost the end of the paragraph.)
3. Throughout the manuscript, there is a conflating of geographically widespread/ locally abundant and geographically restricted/ locally sparse. Generally, in the first part of the paper, the authors use local abundance as a proxy for range size—these two things are not interchangeable and this text needs to be changed. In the latter part of the paper, the authors infer that effects on small vs. large range species are different due to their findings on alpha/ beta/ gamma diversity. This seems more reasonable, but in no cases do they actually measure range size and so claiming their results arise because of range size is problematic. Here are some other examples of this issue: there are a lot of assumptions built into their reasoning in lines 45-47; the linkages described here are really tenuous. Similarly, I do not see how the sentence beginning on line 63 connects to the sentence prior. It seems there is an (unstated) assumption that range size and local abundance are positively correlated (i.e., an abundance occupancy relationship, AORs), but then the authors imply that this AOR is due to dispersal—what about resource use, another hypothesized driver of AORs?

Minor comments:

1. In the abstract: "This is a common..."—I'm not sure exactly what idea the authors are referring to.
2. I find Figures 1B-E quite unhelpful. Perhaps with some more in-text description they could be helpful, but I think Figure 1A is sufficient for this purpose (and I found that the in-text description of Figures 1B-E made that section too long in the introduction anyway).
3. For the other figures, the third axis and the "floating" text in the figure is confusing. Can you put "high delta beta" under differentiation (and similar for homogenization) and remove the delta beta axis? And instead of the floating text, could you add a colored background to show each section of the graph (similar to Figure 1)? I would also put the 1:1 line as the diagonal (to mimic Figure 1).
4. Line 134: I'm not sure what "feedback on the plants" will be, nor what the scale of the "population" is (is that in a block? Plot?)
5. The paragraphs in the results/ discussion are much too long.
6. Lack of clarity with the experimental design: Line 119—sites, plots, and blocks are not defined—definitely need to define at least sites here. There is information missing about the experimental design—was nitrogen alone added? Similarly, line 123: change "after adding nutrients" to "after nutrient additions begun."
7. I don't understand what beta diversity really represents in terms of plots and blocks here (i.e., its biological meaning)—perhaps a concise explanation could help. This seems important given that this metric is critical for the authors' main finding.
8. What is delta r in line 187? I wonder if this didn't get transferred over: There are some symbols that did not get transferred over in the supp info.
9. I'm not sure what to make of the among-site variation in diversity changes, and I'm not sure why the authors are highlighting this.

(Remarks on code availability)

I did not go through the code with a fine tooth comb, but it looks like a usable resource for the community, with reproducible results.

Reviewer #3

(Remarks to the Author)

The manuscript by Chen et al. entitled "Local nutrient addition drives plant biodiversity losses but not biotic homogenisation in global grasslands" summarises the results of the extraordinary global nutrient manipulation experiment. The authors worked in a network of scientists managing 72 experimental sites on six continents. They found that the diversity of both spatially restricted (localised) and widespread species declined at similar rates as a result of nutrient addition. This proportional loss of species at local and larger scales did not lead to significant changes in beta-diversity (across plant origins, i.e. native vs. non-native species and different life forms). Thus, no biotic homogenisation was found, contrary to the authors' expectations and the results of a number of previous studies that found such an effect.

The authors excluded the influence of a number of environmental variables (by including them as covariates in a model) and confirmed that four years was sufficient to obtain reliable results (models including only a subset of the long-term series, i.e. 10 and 14 years, gave similar results) on the observed effects of nutrient addition.

The research addresses a topical issue in contemporary ecology. A better understanding of the patterns and mechanisms of biodiversity change under anthropogenic eutrophication is crucial for predicting future trends and for conservation planning. A novelty of the present research is the broad geographical scope and experimental nature of the research, with a consistent methodology applied over four years at all study sites (14 years at a subset of sites). The scale of the data thus allows results

to be obtained that go beyond the scope of other studies investigating the effects of eutrophication on alpha, beta and gamma diversity of plants.

The hypotheses and assumptions as well as the main results are presented in a clear and interesting way. The language of the manuscript is concise and reads well. However, there are a number of points that have raised my concerns and should be addressed in the revision.

Major remarks

The methods are somewhat chaotically described. The reader has to guess certain measures taken earlier from the description of later parts of the research or from the results section (e.g. conversion of diversity measures to Hill numbers is not mentioned in the section on quantification of diversity metrics). Certain words seem to cause confusion, e.g. line 251, which suggests that micronutrients were only added to a kind of potassium treatment (?), because from the other parts of the manuscript it seems that the authors only focused on plots fertilised with N, P and K together, although the original experimental design described in Borer et al. 2014 also included treatments fertilised with only one macronutrient. I am not sure that the statistical approach used was the most appropriate. Especially the question of modelling and estimating Δ alpha and Δ gamma diversity instead of using the raw data. The nice design of the experiment allows the application of a simple and clear statistical model (ANOVA type), where for each plot one can calculate Δ alpha and Δ beta while for each site Δ gamma diversity and test whether there is an effect of a treatment versus a control for the respective variable. Such an approach also makes it possible to compute confidence intervals for each of the independent variables, allowing the uncertainty of the results to be estimated. The model could include environmental variables, which were treated by the authors as covariates, to remove the portion of variability related to them. One could imagine a mixed effect model where each plot in a site was treated independently (a pair treatment and control in each block), but if they were averaged over all three blocks (as authors did) I see no point in adding a random factor with single observation for each factor level (site). Moreover this averaging of alpha diversity (not Δ alpha) in my opinion removes the effect pairing (before and after) and thus increases uncertainty (averaging over site could be done e.g. after Δ alpha was computed for each of the three plots, which would better include the pairing of the plots). As for the more advanced statistical approach I would rather check the spatial autocorrelation of the model residuals, as the plots are not evenly distributed around the world and one might expect some local patterns to appear (related to an unmeasured factor such as the evolutionary history of local species pools and their adaptation to different nutrient levels), which could be removed e.g. by adding a "region" as a random factor. The log response ratios could also be easily calculated from the raw data. I am not claiming that the Bayesian methods used by the authors were flawed, but I would expect them to be better justified. Perhaps there was a particular reason for using more sophisticated statistical methods based on probability to analyse data from such a simple experimental design, but I would expect such a decision to be thoroughly (with the use of citations) justified in the methods section.

The same applies to the use of Hill numbers rather than raw species richness data. I can understand that the use of Hill numbers can help to linearize otherwise non-linear richness data (in respect to species addition), but the authors did not demonstrate that there was a problem with non-linearity of diversity values and that such a data transformation was necessary.

I mentioned above the potential problem of spatial autocorrelation of model residuals due to unaccounted for local factors affecting community responses to nutrient addition (especially given that experimental sites appear to be clustered in certain biogeographical regions). I would suggest to check whether such spatial autocorrelation is present and can affect the results. An additional comment relates to the calculation of gamma diversity. Due to the phenomenon of distance decay, even under similar environmental conditions, species composition would become less similar as the distance between sampled communities increased. The distances between blocks at each site varied widely (from metres to kilometres), which could have affected gamma diversity (larger gamma diversity if blocks were located far away from each other) and thus beta diversity (which was simply calculated as gamma/alpha diversity). Of course, if we account for change over time (so Δ gamma and Δ beta) this becomes less important, but perhaps distance between blocks should be included as a covariate in a model where gamma diversity is present. The large distance between blocks can also lead to the larger habitat heterogeneity within each site. If environmental conditions (incl. management intensity) included as covariates in the model were averaged per site this can lead to larger bias of results. Of course this could be minimised if each block was analysed separately and site added a random factor.

Number of observations are missing in the figures' captions. Please add it there (or directly on the figures if 'n' differed among the plots).

Other remarks

Abstract:

The 4-year duration of the main experiment should be mentioned.

Lines 32 and 48: "plant origins" is an ambiguous term and should probably be changed into "between native and not-native species".

Lines 123, 341 and 342: The wording ("after adding", "post treatment") suggests one-time nutrient addition while in lines 120 and 261 authors clearly state the nutrients were added annually. I would also include information when the nutrients were applied in relation to the most intensive growing period at each of the sites (before, after).

Line 140: I understand that the distinction between localized and widespread species was based on their presence in only single plot one versus more plots of the same site. This could be a method of distinction between rare and widespread species but at the same time "localised" species could be otherwise widespread and generalist species from other communities (e.g. forest species in grassland plots). Authors could discuss this issue better.

Lines 122-123 (also in other lines): numbers up to nine should be spelled out if it is not a measurement or it is not justified by the other information (e.g. on grouping or units) in the sentence.

Line 172: I would suggest to start a new paragraph

Line 187: typo in 'delta-r'

Line 266: What is "average minimum distance" between three blocks?

Fig. S4. The 95% confidence bands for the linear regression should be drawn, as the non-significance of the relation means that the slope coefficient bands are overlapping with zero (so the relationship could be both positive or negative) not that the diversity change is overlapping with zero...

(Remarks on code availability)

I was able to install and run the code. I found it easy to follow and the results seem to be reproducible with the code.

Version 1:

Reviewer comments:

Reviewer #2

(Remarks to the Author)

I think the manuscript is much improved and easier to understand. I do have a number of comments and questions—the vast majority of which are additional issues that have appeared with the edits included in this submission (i.e., arising from text that was changed since the first submission).

Line 102: change decline to "declines" or similar

Line 103: change it to "nutrient enrichment"

Line 104: greater than what? I think you should say large instead

Line 105: unlike the other reviewers, I do not like the word topology because it has a specific mathematical meaning. What about "framework" ?

Line 108: add "similar" before "proportional"

Line 108: you do not measure how spatially restricted plants are or how widespread they are—you only measure local occupancy and then assume that range size correlates with those. Please use other words such as "common" or "rare" (which you can then define later)

Line 136: what kind of scope are you talking about? Spatial extent? Please be explicit. I suppose the studies referenced in lines 136-139 suffer from these problems. If so, I would change "while some" to "these short term, spatially restricted"

Line 160: so I think this should read three or more blocks? As written it sounds like it is sometimes less than 3.

Line 169-170: the use of "spatially restricted" is still not right here—you did not measure whether spatially restricted species or widespread species were lost, you used the diversity values to INFER that spatially restricted species or widespread species were lost (right?). please revise.

Line 196: 1 should be one

Line 198: was should be were

Lines 201: change "rare" to "locally uncommon"

Lines 202: change "common" to "locally abundant", "relatively rare and common" to locally uncommon and locally abundant". Then if you want to say that locally uncommon species tend to have small geographic ranges, and vice versa, that seems ok.

Line 202: change "this" to "this result"

Line 216: patterns of response to nutrient addition? Also, change this to "this result"

Line 218: change "this" to "this possibility"

Line 227: I do think the results would be improved, here and throughout, by adding in explanations for what these results mean. For example, here at line 227, the authors could say, suggesting a slight, nonsignificant, loss of widespread species. The authors could take out the text on lines 238-241 and perhaps 245-246 (which do not address biotic homogenization) to make space.

Line 242: change "to" to "by"

Line 287: change "less sensitive to rare species than species richness" "that incorporate evenness"

Line 307: Nutnet database?

Line 385: shouldn't Euclidean be capitazlied?

Line 386, 387: change "distance" to "distance among blocks"

(Remarks on code availability)

Reviewer #3

(Remarks to the Author)

The Authors have thoroughly addressed all the comments made in the first revision. The clarity of the methodological section and the terminology used have been significantly improved. I found only a few minor points that need to be corrected:

- Line 117: I would suggest to reformulate this description to 'species with low frequency', given that the cover is not directly related to rarity (small species can be very abundant, yet still cover a small area).
- Line 208: The sentence is unclear (there seems to be some repetition in it).

- Line 317: Here it is stated that the cover was recorded before the addition of nutrients, whereas in line 302 you wrote that the nutrients were added before the growing season. Please clarify this point.
- Fig. S2: I have the impression that the thick lines are not black but blue.
- Fig. S3: Please put "(51)" in the first and third row of plots on the same level as all other numbers. The units of block distance and site productivity should be added.
- Table S8: it is not clear what the difference is between "x" and "X". Some of the entries remain unclear and need to be corrected (e.g. "I can help if needed").
- References: Style of the list should be corrected to the journal style (e.g. journals names in abbreviated format)
- References: 19 - a typo (1?) in the publication title
- References: 23 - year of the publication should be placed in the end

(Remarks on code availability)

First of all, we thank the reviewers and editors for the constructive comments and suggestions that, we believe, have strengthened our manuscript. We considered all suggestions and comments very seriously and revised our manuscript accordingly. Below, we give a detailed point-by-point response to the reviews. To clarify how and where we dealt with the suggestions, we added our responses (with numbering) right after the comments and suggestions (text in blue), and the actual changed text in the manuscript is indicated in red text, with their corresponding page and line numbers in the revised version.

Reviewer #1 (Remarks to the Author):

The experimental study of diversity loss in plant communities in response to fertilization is not new. The focus on how diversity may change in response to fertilization at multiple spatial scales, alter beta-diversity, and potentially affect biotic homogenization is more recent but is addressed by a fair amount of previous work. The more novel contribution that this paper makes is the application to the Nutrient Network Data of a new conceptual framework/topology to explore how different kinds of scale-dependent diversity changes may affect biotic homogenization/differentiation (recently published in Science Advances paper; Blowes et al. 2024).

The framework/topology of Blowes et al. (2024) provides a powerful tool for exploring how plant diversity losses, observed in fertilization studies such as NutNet, may vary in magnitude with spatial scale (alpha vs. gamma). The framework provides a robust means to test the hypothesis that enrichment will lead to compositional homogenization through the loss of localized species that exceed the loss of widespread species.

1. We are pleased to hear that you find our work novel in applying this conceptual framework/topology to the Nutrient Network Data to explore scale-dependent diversity change. We emphasize these points in the abstract and main text. Page 3, line 105-107; page 10, line 282-284. “We apply a topology linking changes in species richness across scales that characterizes changes in the numbers of spatially restricted and widespread species to a standardized experiment adding nutrients across 72 grasslands on six continents.”

“To summarize, we apply a topology that characterizes changes in the number of spatially restricted and widespread species to a globally distributed long-term experiment to provide a comprehensive synthesis of the impact of nutrient addition on scale-dependent plant diversity change in grasslands.”

In this study, evaluation of relative changes in alpha and gamma diversity in response to experimental nutrient enrichment (fertilized versus control plots) showed no evidence for biotic homogenization or differentiation, indicating that the magnitude of species losses to fertilization for localized and widespread species were similar across the 72 sites. This is an important result and runs at odds to common expectation for plant communities.

2. We are pleased to hear that you agree with the importance of our finding.

Is this study original? Yes and no. Yes because it is the first study I know of that provides a robust general test of the homogenization hypothesis for plant communities exposed to nutrient enrichment. No because the Blowes et al paper (2024), which has some of the same authors, develop the framework and applied it in a somewhat similar manner to test for homogenization/differentiation using temporal change data from 461 metacommunities.

3. Indeed, our work builds on the framework and analyses of the Blowes et al paper (2024) (of which the first and senior authors of this paper were the main contributors). Our work also expands beyond this first framework in the following ways: 1) Foremost, our analyses used the framework, but applied it to a globally-distributed experiment where a global change factor (nutrient addition) was directly manipulated to test specific hypotheses, for instance, nutrient addition would cause

biotic homogenization. 2) As a result, our data were consistent and directly comparable. 3) We also investigated whether patterns of diversity changes were consistent across groups of species with similar characteristics (e.g., native and non-native species, life forms).

On balance though I think this manuscript can be an important paper with wide potential interest and impact because of its focus on advancing understanding of mechanisms underlying scale-dependent plant diversity change under eutrophication.

Overall, the manuscript was well-written and clear in presentation. I could not find typos or significant grammatical problems. The basic analytical approach/philosophy is sound, and the statistical modeling seems robust, although some aspects are a bit beyond my knowledge to judge. I appreciate the effort to explore the potential site differences with covariates, analyses to explore potential differences among functional groups, and the sensitivity analyses. I see no flaws in interpretation or conclusions. Overall, a good paper worthy of publication.

4. Thank you.

Reviewer #2 (Remarks to the Author):

This paper quantifies how nutrient addition affects beta diversity, including discussion of the implications of their findings for biotic homogenization. The authors build on previous work showing that both rare and common species are extirpated due to nutrient addition. Namely, the authors show a similar magnitude of reduction in gamma vs. alpha diversity declines, suggesting that beta diversity does not change (and thus biotic homogenization is not occurring). I have a number of concerns, as well as some minor issues.

5. Thank you very much for raising critical questions, and offering many constructive suggestions to improve our manuscript.

Major issues:

1. The authors could improve the connection between beta diversity and biotic homogenization. The paper's introduction is framed around the idea of biotic homogenization, but Figure 1 and the vast majority of the analyses are based on beta diversity, meaning that the connection with biotic homogenization is at times unclear.

6. There are many different definitions for both β diversity and biotic homogenization. Here, we use a standard definition of biotic homogenization as a decrease in β diversity. We clarified the concept of homogenization and its connection to beta diversity in the introduction and methods. Page 4, line 130-132; Page 11, line 334-337; Page 11, line 341-342.

“It is often assumed that nutrient enrichment, like many other global changes, results in biotic homogenization (i.e., increasing similarity in composition among local communities, quantified as a decrease in β diversity)¹⁷⁻²¹. ”

“The typology relies on Whittaker's multiplicative β diversity partition, it measures changes in the effective number of communities¹². As such, if all subplots share the same species, then β diversity would equal to one. In contrast, if each subplot has unique species, then β diversity would equal to three.”

“A decrease in $\Delta\beta$ indicates nutrient addition causes species composition to be more similar among three subplots than that among control subplots.”

2. Further, the focus in the paper is on losses of range-restricted species (where they assume that locally sparse species are range-restricted), but most treatments of biotic homogenization show that invasive species are a key driver. I see that the authors try to say early on that invasive species are

not the key driver in this case (in lines 84-86), but this connection in those sentences is not at all explicit.

7. We agree that non-native/invasive species can be a key driver to aspects of biotic homogenization. However, the other way that biotic homogenization is often conceptualized is through losses of rarer species. Indeed, our focus is to examine the effects of nutrient addition on plant diversity across spatial scales (α - γ - β diversity) through changes in the number of spatially restricted and widespread species. Although we were not able to test whether non-native/invasive species are more linked to widespread species in this study, we tested scale-dependent diversity change for native and non-native species separately. We found that, compared to non-native species, nutrient addition resulted in a 9% greater reduction in α diversity and a 16% greater reduction in γ diversity of native plants, respectively. Also, the overall and site-level patterns of plant diversity changes across spatial scales for native species was more similar to that of the entire communities. Our results suggest that community-level diversity changes were mainly driven by changes in native species.

Further, because biotic homogenization includes changes in both invasive and native plants, it's not at all clear to me how the analyses described in the paragraph beginning on line 157 connect with biotic homogenization. I think there needs to be some clarification of hypotheses here, much earlier in the paragraph. (e.g., "If invasive species are driving biotic homogenization, beta diversity should..."). (note that I would suggest the same for the treatment of life forms, where again the hypotheses are not clear until almost the end of the paragraph.)

8. Using the framework developed by (Blowes *et al.* 2024), we investigated changes in α , γ , and β diversity of the entire communities through changes in the numbers of spatially restricted and widespread species. We also investigated whether patterns of plant diversity changes were consistent across groups of species (e.g., native and non-native species, different life forms). We clarified why we looked into different groups of species. Page 7, line 215-219; Page 8, line 237-238

"The overall proportional species loss within the community at local and larger scales may result if different species groups have contrasting patterns. For instance, this could be the case if native species loss is greater at the larger spatial scale than at the local scale, while non-native species loss is lower at the larger than the local scale. To test this, we investigate changes in α , γ , and β diversity for native and non-native species separately."

"We also separated species into graminoid, forb, legume, and woody species to investigate scale-dependent diversity change."

3. Throughout the manuscript, there is a conflating of geographically widespread/ locally abundant and geographically restricted/ locally sparse. Generally, in the first part of the paper, the authors use local abundance as a proxy for range size—these two things are not interchangeable and this text needs to be changed.

9. We agree that geographical range distribution and abundance are two distinct categories. In terms of range distribution, which we define within a site, species can be categorized into widespread and spatially restricted species. In terms of abundance, species can be categorized into rare, common, and abundant species. Rare species is not equal to spatially restricted species, likewise, abundant species is not equal to widespread species. We thoroughly checked our text, made clearer our definitions, and deleted any speculations linking the two categories. For instance, Page 4, line 116-118

"This diversity decline is typically ascribed to disproportionate losses of rare species (i.e., species with low cover) because small populations are more susceptible to extinction⁶⁻⁸. In addition, nutrient enrichment often leads to the removal of species with traits ill-suited for effectively competing in high nutrient conditions⁶."

In the latter part of the paper, the authors infer that effects on small vs. large range species are different due to their findings on alpha/ beta/ gamma diversity. This seems more reasonable, but in no cases do they actually measure range size and so claiming their results arise because of range size is problematic.

10. Indeed, we did not measure species' geographic range size. We revised the text to make it clearer that we are referring to occupancy (i.e., whether species occupy many or only a few subplots within sites). Also, we acknowledged this limitation in the discussion. Page 3, line 107-109; Page 10, line 276-280

“Overall, we found similar declines of spatially restricted and widespread species, and proportional species loss at local and larger scales, and no biotic homogenization after 4 years and up to 14 years of treatment. ”

“Also, the scale in which we inferred changes in the number of spatially restricted and widespread species, by examining how many local communities they were lost from, is on a small scale. Linking estimates of species' geographic range size and their traits with changes in plant diversity across spatial scales^{19,42} will deepen our understanding of the mechanisms of diversity change under nutrient enrichment. ”

Here are some other examples of this issue: there are a lot of assumptions built into their reasoning in lines 45-47; the linkages described here are really tenuous.

11. We agree with you that the argument was weak in this sentence, and we deleted it.

Similarly, I do not see how the sentence beginning on line 63 connects to the sentence prior. It seems there is an (unstated) assumption that range size and local abundance are positively correlated (i.e., an abundance occupancy relationship, AORs), but then the authors imply that this AOR is due to dispersal—what about resource use, another hypothesized driver of AORs?

12. We agree with you that the sentence did not connect well to the sentence prior, it was not essential and to avoid confusion, we have now deleted it.

Minor comments:

1. In the abstract: “This is a common...”—I'm not sure exactly what idea the authors are referring to.

13. We agree this is confusing. We have revised the text. Page 3, line 102-105

“A common but untested expectation is that it reduces variation in nutrient conditions among localities and selects for a smaller pool of species, causing greater diversity declines at larger scales and biotic homogenization.”

2. I find Figures 1B-E quite unhelpful. Perhaps with some more in-text description they could be helpful, but I think Figure 1A is sufficient for this purpose (and I found that the in-text description of Figures 1B-E made that section too long in the introduction anyway).

14. As you suggested, we deleted Figures 1B-E.

3. For the other figures, the third axis and the “floating” text in the figure is confusing. Can you put “high delta beta” under differentiation (and similar for homogenization) and remove the delta beta axis? And instead of the floating text, could you add a colored background to show each section of the graph (similar to Figure 1)? I would also put the 1:1 line as the diagonal (to mimic Figure 1).

15. Thank you for this great suggestion. We adjusted figures as suggested.

4. Line 134: I'm not sure what "feedback on the plants" will be, nor what the scale of the "population" is (is that in a block? Plot?)

16. We deleted this sentence, and clarified clearer what we meant. Page 9, line 273-276

"Despite our evidence for little change in β diversity under nutrient addition across the entire experiment, we recognize limitations of extrapolating these results to the landscape scale (e.g., $>1 \text{ km} \times 1 \text{ km}$). This is because many ecological processes that directly and indirectly influence plant diversity can be very different at the landscape scales^{14,42}."

5. The paragraphs in the results/ discussion are much too long.

17. We substantially revised and condensed the Results and Discussion.

6. Lack of clarity with the experiential design: Line 119—sites, plots, and blocks are not defined—definitely need to define at least sites here. There is information missing about the experimental design—was nitrogen alone added?

18. Thank you for bringing this to our attention. Nitrogen was not added alone. Nitrogen, phosphorus, and potassium were added together and annually. We now explain in detail the experimental design, sites, plots, and blocks at the end of the introduction. Page 4, line 155-169
"Here, we use this topology to synthesize scale-dependent plant diversity change (for the entire community and groups of species) under nutrient enrichment using a long-term standardized experiment in 72 grasslands distributed across six continents (i.e., NutNet³⁹; Fig. S1; Table S1). We used two treatments: Ambient (Control) and fertilization by nitrogen, phosphorus, and potassium together (i.e., NPK). Nutrients were added at a rate of 10 g m^{-2} annually. Treatments were randomly assigned to $5 \text{ m} \times 5 \text{ m}$ plots and were replicated in three blocks at most sites. Species composition was sampled in one $1 \text{ m} \times 1 \text{ m}$ permanent subplot using a standardized protocol. At each site, α diversity was determined as the number of species in each permanent subplot (i.e., species richness), and γ diversity as the total number of species occurring over three permanent subplots (for each treatment separately). We excluded additional blocks from sites that had more than three because γ and β diversity depends on the number of local communities used. We calculated $\Delta\alpha$ as the richness difference in local communities (subplots) and $\Delta\gamma$ as the difference in the sum of the subplots under nutrient addition relative to that of control on the log scale. That is, $\Delta\alpha = \log(\alpha_{\text{NPK}}/\alpha_{\text{Control}})$ and $\Delta\gamma = \log(\gamma_{\text{NPK}}/\gamma_{\text{Control}})$. We then calculated $\Delta\beta$ as $\Delta\gamma$ minus $\overline{\Delta\alpha}$, where $\overline{\Delta\alpha}$ is the average of $\Delta\alpha$ over three blocks."

Similarly, line 123: change "after adding nutrients" to "after nutrient additions begun."

19. We adjusted the text as suggested. Page 4, line 169-172

"In all, we found similar magnitudes of loss for spatially restricted and widespread species, and proportional losses of species at local and larger scales, and no clear biotic homogenization or differentiation four years, and even up to 14 years, after nutrient additions began."

7. I don't understand what beta diversity really represents in terms of plots and blocks here (i.e., its biological meaning)—perhaps a concise explanation could help. This seems important given that this metric is critical for the authors' main finding.

20. We clarified the definition of beta diversity. We added a concise explanation for change in beta diversity in the context of plots and blocks in the methods. Page 11, line 334-337; Page 11, line 341-342.

"The typology relies on Whittaker's multiplicative β diversity partition, it measures changes in the effective number of communities¹². As such, if all subplots share the same species, then β diversity would equal to one. In contrast, if each subplot has unique species, then β diversity would equal to three."

"A decrease in $\Delta\beta$ indicates nutrient addition causes species composition to be more similar among

three subplots than that among control subplots.”

8. What is delta r in line 187? I wonder if this didn't get transferred over: There are some symbols that did not get transferred over in the supp info.

21. This was a typo. We meant $\Delta\gamma$. We deleted this sentence to make the result and discussion section more concise. We also carefully checked the symbols in the supplementary file to make them consistent to the main text.

9. I'm not sure what to make of the among-site variation in diversity changes, and I'm not sure why the authors are highlighting this.

22. This text was deleted because we substantially revised our models as suggested by reviewer 3.

Reviewer #2 (Remarks on code availability):

I did not go through the code with a fine tooth comb, but it looks like a usable resource for the community, with reproducible results.

23. We are glad that you found the code usable and the results reproducible.

Reviewer #3 (Remarks to the Author):

The manuscript by Chen et al. entitled "Local nutrient addition drives plant biodiversity losses but not biotic homogenisation in global grasslands" summarises the results of the extraordinary global nutrient manipulation experiment. The authors worked in a network of scientists managing 72 experimental sites on six continents. They found that the diversity of both spatially restricted (localised) and widespread species declined at similar rates as a result of nutrient addition. This proportional loss of species at local and larger scales did not lead to significant changes in beta-diversity (across plant origins, i.e. native vs. non-native species and different life forms). Thus, no biotic homogenisation was found, contrary to the authors' expectations and the results of a number of previous studies that found such an effect.

The authors excluded the influence of a number of environmental variables (by including them as covariates in a model) and confirmed that four years was sufficient to obtain reliable results (models including only a subset of the long-term series, i.e. 10 and 14 years, gave similar results) on the observed effects of nutrient addition.

The research addresses a topical issue in contemporary ecology. A better understanding of the patterns and mechanisms of biodiversity change under anthropogenic eutrophication is crucial for predicting future trends and for conservation planning.

A novelty of the present research is the broad geographical scope and experimental nature of the research, with a consistent methodology applied over four years at all study sites (14 years at a subset of sites). The scale of the data thus allows results to be obtained that go beyond the scope of other studies investigating the effects of eutrophication on alpha, beta and gamma diversity of plants.

The hypotheses and assumptions as well as the main results are presented in a clear and interesting way. The language of the manuscript is concise and reads well. However, there are a number of points that have raised my concerns and should be addressed in the revision.

24. We appreciate that you value our work and offer many suggestions to improve our analyses and manuscript.

Major remarks

The methods are somewhat chaotically described. The reader has to guess certain measures taken earlier from the description of later parts of the research or from the results section (e.g. conversion of diversity measures to Hill numbers is not mentioned in the section on quantification of diversity metrics).

25. Thank you for pointing that out. We have substantially revised the Methods section. For instance, we explain clearly why and how different diversity measures were used, how they connect to Hill numbers. Moreover, we now explain more clearly the experimental design, sites, plots, and blocks at the end of the introduction and in the methods. Please also see our response 18. Page 11, line 359-367; page 10, line 294-305

“To examine whether diversity change were sensitive to species relative covers, we redid the above analyses (i.e., based on species richness) using Shannon diversity and Simpson diversity (both converted to effective numbers)⁴⁷(Fig. S4). Species richness is most sensitive to rare species, followed by Shannon diversity, and Simpson diversity is more sensitive to the numbers of common (or relatively abundant) species. We calculated the exponential of Shannon diversity and the inverse form of Simpson diversity using the R package *vegan*⁴⁸. These three diversity metrics equal to diversity with order $q = \{0, 1, 2\}$, where increasing q decreases the influence of rare species, and $D_q = \left(\sum_{i=1}^s p_i^q \right)^{1/(1-q)}$, where p is the relative cover of species i , s is the total number of species. These diversity metrics are also referred to as Hill numbers^{47,49}.”

“Experimental setup

The experimental sites used in this study are part of the Nutrient Network (NutNet, Fig. S1 and Table S1). The experimental design includes a factorial manipulation of nutrients (N, P, and K) plus two fences to exclude herbivores, see ref³⁹ for more details. For the analyses here, we used plots under two treatments: Ambient (Control) and fertilization by nitrogen, phosphate, and potassium together (i.e., NPK). A micronutrient mix consists of Fe (15%), S (14%), Mg (1.5%), Mn (2.5%), Cu (1%), Zn (1%), B (0.2%), and Mo (0.05%) was added once only at the start of the experiment (i.e., year 1) for the nutrient addition plots, but not in subsequent years to avoid toxicity. Nitrogen, phosphate, potassium were added annually before the beginning of the growing season at most sites. Nitrogen was added as $10 \text{ g m}^{-2} \text{ yr}^{-1}$ time-release urea $[(\text{NH}_2)_2\text{CO}]$, phosphate was added as $10 \text{ g m}^{-2} \text{ yr}^{-1}$ triple-super phosphate $[\text{Ca}(\text{H}_2\text{PO}_4)_2]$, while potassium was added as $10 \text{ g m}^{-2} \text{ yr}^{-1}$ sulfate $[\text{K}_2\text{SO}_4]$.”

Certain words seem to cause confusion, e.g. line 251, which suggests that micronutrients were only added to a kind of potassium treatment (?), because from the other parts of the manuscript it seems that the authors only focused on plots fertilised with N, P and K together, although the original experimental design described in Borer et al. 2014 also included treatments fertilised with only one macronutrient.

26. Sorry for the confusion. We now describe how nutrients were added more clearly. Specifically, we did not use all of the treatments described in Borer et al. 2014 but only used the plots with all nutrients added compared to the control to minimize complexity. Please also see our response 25.

I am not sure that the statistical approach used was the most appropriate. Especially the question of modelling and estimating $\Delta\alpha$ and $\Delta\gamma$ diversity instead of using the raw data. The nice design of the experiment allows the application of a simple and clear statistical model (ANOVA type), where for each plot one can calculate $\Delta\alpha$ and $\Delta\beta$ while for each site $\Delta\gamma$ diversity and test whether there is an effect of a treatment versus a control for the respective variable. Such an approach also makes it possible to compute confidence intervals for each of the independent variables, allowing the uncertainty of the results to be estimated.

27. We now manually calculate $\Delta\alpha$, $\Delta\gamma$, and $\Delta\beta$ under nutrient addition compared with that of

control treatment at each site. We also present these site-level diversity change data in all figures. As suggested, we now model the $\Delta\alpha$, $\Delta\gamma$, and $\Delta\beta$ as the response variables, and estimate the overall average (i.e., the intercept in the model), and include and site-level variation by including site as a random variable (i.e., varying intercept). Page 4, line 161-169; Page 11, line 347-349

“At each site, α diversity was determined as the number of species in each permanent subplot (i.e., species richness), and γ diversity as the total number of species occurring in three permanent subplots (for each treatment separately). We excluded additional blocks from sites that had more than three because γ and β diversity depend on the number of local communities used. We calculated $\Delta\alpha$ as the richness difference in local communities (subplots) and $\Delta\gamma$ as the difference in the sum of the subplots under nutrient addition relative to that of control on the log scale. That is, $\Delta\alpha = \log(\alpha_{\text{NPK}}/\alpha_{\text{Control}})$ and $\Delta\gamma = \log(\gamma_{\text{NPK}}/\gamma_{\text{Control}})$. We then calculated $\Delta\beta$ as $\Delta\gamma$ minus $\overline{\Delta\alpha}$, where $\overline{\Delta\alpha}$ is the average of $\Delta\alpha$ over three blocks.”

“We fitted multilevel (also referred as mixed effects or hierarchical) models for $\Delta\alpha$, $\Delta\gamma$, and $\Delta\beta$ (as the response variable; all on the log scale) separately. We included random intercept for each site, model was coded as: richness change $\sim 1 + (1 | \text{sites})$ to estimate site-level variation.”

The model could include environmental variables, which were treated by the authors as covariates, to remove the portion of variability related to them. One could imagine a mixed effect model where each plot in a site was treated independently (o a pair treatment and control in each block), but if they were averaged over all three blocks (as authors did) I see no point in adding a random factor with single observation for each factor level (site). Moreover this averaging of alpha diversity (not $\Delta\alpha$) in my opinion removes the effect pairing (before and after) and thus increases uncertainty (averaging over site could be done e.g. after $\Delta\alpha$ was computed for each of the three plots, which would better include the pairing of the plots).

28. Thank you for the suggestions. We agree. We calculated $\Delta\alpha$ as the diversity difference under nutrient addition relative to that of control treatment on the log scale for each of the three blocks. We think including site as a random variable is important, as it allows us to estimate site-level variation. Please also see our response 27.

As for the more advanced statistical approach I would rather check the spatial autocorrelation of the model residuals, as the plots are not evenly distributed around the world and one might expect some local patterns to appear (related to an unmeasured factor such as the evolutionary history of local species pools and their adaptation to different nutrient levels), which could be removed e.g. by adding a "region" as a random factor. The log response ratios could also be easily calculated from the raw data. I am not claiming that the Bayesian methods used by the authors were flawed, but I would expect them to be better justified. Perhaps there was a particular reason for using more sophisticated statistical methods based on probability to analyse data from such an simple experimental design, but I would expect such a decision to be thoroughly (with the use of citations) justified in the methods section.

29. We checked spatial autocorrelation using Moran's I for our response variables. We also checked spatial autocorrelation using the residual data of diversity change estimated from the models, we did not find significant spatial autocorrelation either. Page 11, line 242-345;

“Because sites are not evenly distributed around the world, many sites are aggregated in Norther America, we checked spatial autocorrelation of diversity change under nutrient addition using Moran's I. We found that $\overline{\Delta\alpha}$, $\Delta\gamma$, and $\Delta\beta$ did not appear to be more similar for sites that are closer to each other (Table S2).”

“Table S2.

Summary of spatial autocorrelation for the effects of nutrient addition on alpha, gamma, and beta diversity ($\Delta\alpha$, $\Delta\gamma$, and $\Delta\beta$) four year after nutrient addition began using Moran's I test.

Positive Moran's I suggests similar values cluster together while negative Moran's I suggests dissimilar values are adjacent. The z-score quantifies how extreme the observed Moran's I value is compared with that is expected under the null hypothesis of spatial randomness.

Diversity facets	Moran's I	Standard deviate (z score)	p
$\Delta\alpha$	0.0109	0.2855	0.3876
$\Delta\gamma$	0.0155	0.3429	0.3658
$\Delta\beta$	-0.0306	-0.2007	0.5795

”

We think including site as a random variable is important, it allows us to estimate site-level variation. Please see our reasoning in our response 27 and 28. Since we did not find significant spatial autocorrelation using the diversity data, adding a "region" as a random factor may not be needed.

“We used Bayesian analysis because it yields full posterior distributions of parameters rather than point estimates and p-values, which provides a deeper understanding of the uncertainty and variability in the results⁴⁴.”

The same applies to the use of Hill numbers rather than raw species richness data. I can understand that the use of Hill numbers can help to linearize otherwise non-linear richness data (in respect to species addition), but the authors did not demonstrate that there was a problem with non-linearity of diversity values and that such a data transformation was necessary.

30. We have clarified that we examined the effective number of species based on Shannon and Simpson diversity (in addition to species richness) to examine whether our results were sensitive to species' relative abundances. The typology relies on Whittaker's multiplicative β diversity partition, it measures changes in the effective number of species. Hill numbers with $q = 0$ equals to species richness, Hill numbers with $q = 1$ equals to effective numbers based on Shannon diversity, and Hill numbers with $q = 2$ equals to effective numbers based on Simpson diversity. Page 11, line 359-367

“To examine whether diversity change were sensitive to species relative covers, we redid the above analyses (i.e., based on species richness) using Shannon diversity and Simpson diversity (both converted to effective numbers)⁴⁷(Fig. S4). Species richness is most sensitive to rare species, followed by Shannon diversity, and Simpson diversity is more sensitive to the numbers of common (or relatively abundant) species. We calculated the exponential of Shannon entropy and the inverse form of Simpson diversity using the R package vegan⁴⁸. These three diversity metrics are equal to diversity with order $q = \{0, 1, 2\}$, where increasing q decreases the influence of rare species, and $D_q = \left(\sum_{i=1}^s p_i^q \right)^{1/(1-q)}$, where p is the relative abundance (cover) of species i , s is the total number of species. These diversity metrics are also referred to as Hill numbers^{47,49}. ”

I mentioned above the potential problem of spatial autocorrelation of model residuals due to unaccounted for local factors affecting community responses to nutrient addition (especially given that experimental sites appear to be clustered in certain biogeographical regions). I would suggest to check whether such spatial autocorrelation is present and can affect the results.

31. We checked for spatial autocorrelation. Please see our response 29.

An additional comment relates to the calculation of gamma diversity. Due to the phenomenon of distance decay, even under similar environmental conditions, species composition would become less similar as the distance between sampled communities increased. The distances between blocks at each site varied widely (from metres to kilometres), which could have affected gamma diversity (larger gamma diversity if blocks were located far away from each other) and thus beta diversity (which was simply calculated as gamma/alpha diversity). Of course, if we account for change over

time (so $\Delta\gamma$ and $\Delta\beta$) this becomes less important, but perhaps distance between blocks should be included as a covariate in a model where gamma diversity is present. The large distance between blocks can also lead to the larger habitat heterogeneity within each site. If environmental conditions (incl. management intensity) included as covariates in the model were averaged per site this can lead to larger bias of results. Of course this could be minimised if each block was analysed separately and site added a random factor.

32. $\Delta\gamma$ and $\Delta\beta$ were not calculated as change over time, but as the difference under nutrient addition compared with that under control treatment.

We checked whether distance among blocks impacted our estimate of $\Delta\alpha$, $\Delta\gamma$, and $\Delta\beta$, and we found no strong correlations. Please see Fig. S3. Page 12, line 381-387

“We also investigated relationships between change in diversity and distance among blocks, because species composition may become less similar as the distance between sampled communities increases. The average pairwise distance among the three blocks within sites ranged from 23.04 to 12538.09 m, with a mean of 513.01 m and a median of 118.7 m across 54 sites that have geolocation data for each block. We first calculated three euclidean distances between pairs of blocks, we then used the mean of these pairwise distances as the average distance. We used the average distance instead of area, because blocks are arranged in parallel at some sites. We fitted linear regression models for $\overline{\Delta\alpha}$, $\Delta\gamma$, and $\Delta\beta$ as the response variable separately, and each of the site characteristics was used as a predictor variable.”

Number of observations are missing in the figures' captions. Please add it there (or directly on the figures if 'n' differed among the plots).

33. We added the number of observations directly on the figures as suggested.

Other remarks

Abstract:

The 4-year duration of the main experiment should be mentioned.

34. We added 4-year duration in the abstract. Page 3, line 107-109

“Overall, we found similar declines of spatially restricted and widespread species, and proportional species loss at local and larger scales, and no biotic homogenization after 4 years and up to 14 years of treatment.”

Lines 32 and 48: “plant origins” is an ambiguous term and should probably be changed into “between native and not-native species”.

35. We adjusted this accordingly in the abstract and main text. Page 3, line 109-111; Page 7, line 215-219

“These patterns of diversity changes were generally consistent across species groups (e.g., native species, graminoids and legume species).”

“The overall proportional species loss within the community at local and larger scales may result if different species groups have contrasting patterns. For instance, this could be the case if native species loss is greater at the larger spatial scale than at the local scale, while non-native species loss is lower at the larger than the local scale. To test this, we investigate changes in α , γ , and β diversity for native and non-native species separately.”

Lines 123, 341 and 342: The wording (“after adding”, “post treatment”) suggests one-time nutrient addition while in lines 120 and 261 authors clearly state the nutrients were added annually. I would also include information when the nutrients were applied in relation to the most intensive growing period at each of the sites (before, after).

36. We checked this wording throughout the text to be consistent that nutrients were applied

annually. We now explain in detail the experimental design, sites, plots, and blocks at the end of the introduction. Please see our response 18 and 25. We clarified the text including information when the nutrients were applied. Page 10, line 301-303

“Nitrogen, phosphate, potassium were added annually before the beginning of the growing season at most sites.”

Line 140: I understand that the distinction between localized and widespread species was based on their presence in only single plot one versus more plots of the same site. This could be a method of distinction between rare and widespread species but at the same time “localised” species could be otherwise widespread and generalist species from other communities (e.g. forest species in grassland plots). Authors could discuss this issue better.

37. We agree. We clarified what we meant and did with changes in the number of widespread and spatially restricted species. Page 10, line 276-280

“Also, the scale in which we inferred changes in the number of spatially restricted and widespread species, by examining how many local communities they were lost from, is on a small scale.

Linking estimates of species’ geographic range size and their traits with changes in plant diversity across spatial scales^{19,42} can deepen our understanding of the mechanisms of diversity change under nutrient enrichment.”

Lines 122-123 (also in other lines): numbers up to nine should be spelled out if it is not a measurement or it is not justified by the other information (e.g. on grouping or units) in the sentence.

38. Thank you for bringing this detail to us, we spelled out numbers up to nine when it is not a measurement or it is not justified by the other information. Page 4, line 161-164

“At each site, α diversity was determined as the number of species in each permanent subplot (i.e., species richness), and γ diversity as the total number of species occurring over three permanent subplots (for each treatment separately).”

Line 172: I would suggest to start a new paragraph

39. We adjusted accordingly.

Line 187: typo in ‘delta-r’

40. Thank you for spotting this mistake. We deleted this sentence to make the result and discussion section more concise as the reviewer 2 suggested.

Line 266: What is “average minimum distance” between three blocks?

41. Sorry for the confusion, we have rephrased as below. Page 12, line 383-387

“The average pairwise distance among the three blocks within sites ranged from 23.04 to 12538.09 m, with a mean of 513.01 m and a median of 118.7 m across 54 sites that have geolocation data for each block. We first calculated three euclidean distances between pairs of blocks, we then used the mean of these pairwise distances as the average distance. We used the average distance instead of area, because blocks are arranged in parallel at some sites.”

Fig. S4. The 95% confidence bands for the linear regression should be drawn, as the non-significance of the relation means that the slope coefficient bands are overlapping with zero (so the relationship could be both positive or negative) not that the diversity change is overlapping with zero...

42. We added the 95% confidence bands for Fig. S3 (i.e. the Fig. S4 in the previous version of the text).

Reviewer #3 (Remarks on code availability):

I was able to install and run the code. I found it easy to follow and the results seem to be reproducible with the code.

43. Thank you for checking our R code. We substantially revised the models as you suggested. We calculated $\Delta\alpha$, $\Delta\gamma$, and $\Delta\beta$ directly, and included them in the models as the response variable.

We thank the reviewers and editors for the additional constructive comments and suggestions to strengthen our manuscript. We seriously considered all comments and revised our manuscript accordingly. Below, we give a detailed point-by-point response to the reviews. To clarify how and where we dealt with the suggestions, we added our responses (with numbering) right after the comments and suggestions (text in blue), and the actual changed text in the manuscript is indicated in red text, with their corresponding page and line numbers in the revised version.

Reviewer #2 (Remarks to the Author):

I think the manuscript is much improved and easier to understand. I do have a number of comments and questions—the vast majority of which are additional issues that have appeared with the edits included in this submission (i.e., arising from text that was changed since the first submission).

Line 102: change decline to “declines” or similar

1. We revised as you suggested. Page 3, line 94

“Nutrient enrichment typically causes local plant diversity **declines**.”

Line 103: change it to “nutrient enrichment”

2. We revised. Page 3, line 94-97

“A common but untested expectation is that **nutrient enrichment also** reduces variation in nutrient conditions among localities and selects for a smaller pool of species, causing greater diversity declines at larger **than local scales** and **thus** biotic homogenization.”

Line 104: greater than what? I think you should say large instead

3. We now state “larger than the local scale” for clarity, please see response 2 above. We prefer to use “larger” instead of “large”, and now clarify that “larger” is in relation to the local scale.

Line 105: unlike the other reviewers, I do not like the word topology because it has a specific mathematical meaning. What about “framework” ?

4. We changed “topology” to “framework” throughout the text.

Line 108: add “similar” before “proportional”

5. We revised as you suggested. Page 3, line 99-102

“Overall, we **find** proportionally **similar** species loss at local and larger scales, **suggesting** similar declines of spatially restricted and widespread species, and no biotic homogenization after 4 years and up to 14 years of treatment.”

Line 108: you do not measure how spatially restricted plants are or how widespread they are—you only measure local occupancy and then assume that range size correlates with those. Please use other words such as “common” or “rare” (which you can then define later)

6. You are right. We did not directly measure how spatially restricted plants are or how widespread they are. Instead, we infer changes in occupancy through changes in α and γ diversity with the framework. We do not link rarity (in terms of relative cover) with occupancy (spatially restricted or widespread species), or range size, which is beyond the scope of our study. The terms “common” or “rare” may be misleading as well. Because most researchers would intuitively link them to species abundance. We revised the text in the abstract to reflect that changes in the number of spatially restricted and widespread species were inferred. Page 3, line 97-99; line 99-102

“Here we apply a framework that links changes in species richness across scales to changes in the numbers of spatially restricted and widespread species for a standardized nutrient addition experiment across 72 grasslands on six continents.”

“Overall, we find proportionally similar species loss at local and larger scales, suggesting similar declines of spatially restricted and widespread species, and no biotic homogenization after 4 years and up to 14 years of treatment. ”

Line 136: what kind of scope are you talking about? Spatial extent? Please be explicit. I suppose the studies referenced in lines 136-139 suffer from these problems. If so, I would change “while some” to “these short term, spatially restricted”

7. We revised as you suggested. Page 4, line 130-134

“Previous investigations of scale-dependent diversity change under nutrient enrichment have tended to be short term or limited in spatial extent²⁴⁻²⁸. These short-term or spatially-restricted studies have found mixed results, indicating that nutrient enrichment leads to biotic homogenization²⁹⁻³¹, no changes in β diversity^{24,26,27,32} or even differentiation (i.e., increase in β diversity)^{25,28,33-37}. ”

Line 160: so I think this should read three or more blocks? As written it sounds like it is sometimes less than 3.

8. We revised as you suggested. Page 4, line 154-155

“Treatments were randomly assigned to 5 m × 5 m plots and were replicated in three or more blocks. ”

Line 169-170: the use of “spatially restricted” is still not right here—you did not measure whether spatially restricted species or widespread species were lost, you used the diversity values to INFER that spatially restricted species or widespread species were lost (right?). please revise.

9. We agree with you. We did not measure how spatially restricted plants are or how widespread they are. We infer it through changes in α and γ diversity using the framework. We revised the abstract and text here to reflect that. Please see our response 6 and page 5, line 164-165

“Overall, we find proportionally similar species loss at local and larger scales, suggesting similar magnitudes of declines of spatially restricted and widespread species. ”

Line 196: 1 should be one

10. We revised as you suggested. Page 5, line 177-179

“On a site level, we found biotic homogenization at 24 sites, differentiation at 47 sites, and no change in β diversity at one site. ”

Line 198: was should be were

11. We revised as you suggested. Page 5, line 181

“Importantly, the overall effects of nutrient addition on α , γ , and β diversity were similar ”

Lines 201: change “rare” to “locally uncommon”

12. We prefer to use rare as a measure for “low relative species cover” as we defined in the beginning of the introduction. Page 4, line 111-112

“This diversity decline is typically ascribed to disproportionate losses of rare species (i.e., species with relatively low cover) ”

Lines 202: change “common” to “locally abundant”, “relatively rare and common” to locally uncommon and locally abundant”. Then if you want to say that locally uncommon species tend to have small geographic ranges, and vice versa, that seems ok.

13. We prefer to use rare and abundant as a measure for “relative species cover”. We do not link rarity (in terms of relative cover) with occupancy (spatially restricted or widespread species), or range size, which is beyond the scope of our study. Please see our response 12.

Line 202: change “this” to “this result”

14. We revised as you suggested. Page 5, line 185-186

“Because species richness is more strongly influenced by rare species, while Shannon and Simpson diversity increasingly weigh **abundant** species, this **result** suggests that relatively rare and **abundant** species responded similarly to nutrient addition.”

Line 216: patterns of response to nutrient addition? Also, change this to “this result”

15. We revised as you suggested. Page 5, line 189-190; line 191

“The overall proportional species loss within the community at local and larger scales **on average** may result if different species groups have contrasting patterns **of response to nutrient addition.**”
“For instance, this **result** could be the case ”

Line 218: change “this” to “this possibility”

16. We revised as you suggested. Page 5, line 192-193

“To test this **possibility,**”

Line 227: I do think the results would be improved, here and throughout, by adding in explanations for what these results mean. For example, here at line 227, the authors could say, suggesting a slight, nonsignificant, loss of widespread species. The authors could take out the text on lines 238-241 and perhaps 245-246 (which do not address biotic homogenization) to make space.

16. We revised as you suggested by adding explanations for what the results mean. We prefer to use a strict cut off (95% credible interval) to define significant effects, and we focus on describe the significant effects and their meanings. Page 6, line 213-215.

“**Nutrient addition caused a weak biotic homogenization for woody species ($\Delta\beta = -0.14$; 95% credible interval: -0.30 to 0.003 ; Table S6), this was primarily linked to loss of spatially restricted species (Fig. 4D).** ”

We deleted the text from lines 238-241. We retained the text from lines 245-246 because they describe how α and γ diversity change, which is tightly linked to β diversity within species groups.

To make space, reduce repetition, and make things easier for readers, we simplified the robustness test by keeping 1) using a subset of 16 sites that had data 14 years after nutrient additions began and 2) using 11 sites that had five spatial blocks. That said, we deleted 1) a subset of 29 sites that had data 10 years after nutrient additions began and 2) using 16 sites that had four spatial blocks.

Line 242: change “to” to “by”

17. We revised as you suggested. Page 5, line 206-207

“Similar to that of entire communities, nutrient addition decreased α and γ diversity **by** similar magnitudes”

Line 287: change “less sensitive to rare species than species richness” “that incorporate evenness”

18. We changed to “incorporate species relative cover” because it is more intuitive and we did not use or define “evenness” in our study. Page 6, line 244-246

“These overall patterns were largely consistent for diversity metrics that **incorporate relative species covers**, across species groups, and over long time periods.”

Line 307: Nutnet database?

19. We revised as you suggested. Page 6, line 265

“Data were retrieved from the NutNet **database** in November 2023. ”

Line 385: shouldn't Euclidean be capitazlied?

20. Yes, it shoud. Thanks for spotting this typo. Page 8, line 343

“We first calculated three **E**uclidean distances”

Line 386, 387: change “distance” to “distance among blocks”

21. We revised as you suggested. Page 8, line 344-345

“ we then used the mean of these pairwise distances as the average distance **among blocks**. We used the average **distance among blocks** instead of area”

Reviewer #3 (Remarks to the Author):

The Authors have thoroughly addressed all the comments made in the first revision. The clarity of the methodological section and the terminology used have been significantly improved. I found only a few minor points that need to be corrected:

- Line 117: I would suggest to reformulate this description to 'species with low frequency', given that the cover is not directly related to rarity (small species can be very abundant, yet still cover a small area).

22. Rarity can broadly describe geographical range, habitat specificity, or local abundance. We prefer to use rare, common, abundant as a measure of relative species cover. We defined this clearly in the beginning of the introduction. We do not link rarity (in terms of relative cover) with occupancy (spatially restricted or widespread species), or range size, which is beyond the scope of our study. Page 4, line 111-112

“This diversity decline is typically ascribed to disproportionate losses of rare species (i.e., species with **relatively** low cover) ”

- Line 208: The sentence is unclear (there seems to be some repetition in it).

23. Thanks for spotting the repetition. We revised. Page 12, line 430-431

“When a site has $\overline{\Delta\alpha} = 0$, $\Delta\gamma = 0$, or $\Delta\beta = 0$, it was not counted into any of the six scenarios **as shown in the framework**. ”

- Line 317: Here it is stated that the cover was recorded before the addition of nutrients, whereas in line 302 you wrote that the nutrients were added before the growing season. Please clarify this point.

24. Thanks for spotting this inconsistency. According to the protocol of NutNet (<https://nutnet.org/nutrients>), nutrients were added before the growing season of each treatment year. We revised the text to make our description consistent to the protocol of NutNet. Page 6, line 260-261; Page 7, line 274-275

“Nitrogen, phosphate, potassium were added annually **before the growing season of each treatment year** at most sites. ”

“At most sites, cover was recorded once per year at peak biomass.”

- Fig. S2: I have the impression that the thick lines are not black but blue.

25. You are right. We revised the text.

“Predicted values (the thin lines) and observed values (the thick lines) for change in alpha, gamma, and beta diversity ($\Delta\alpha$, $\Delta\gamma$, and $\Delta\beta$) under nutrient addition.”

- Fig. S3: Please put "(51)" in the first and third row of plots on the same level as all other numbers. The units of block distance and site productivity should be added.

26. We revised the Fig. S3 as you suggested. We added the units of block distance and site productivity in the caption because there is no enough space in the axis x.

“Unit for block distance is meter, unit for site productivity is g m^{-2} . ”

- Table S8: it is not clear what the difference is between "x" and "X". Some of the entries remain unclear and need to be corrected (e.g. "I can help if needed").

27. We corrected the authorship contribution table. We deleted all extraneous entries. All “X” are replaced with “x”, and the contributions for each author is marked with x. We explained this in the table caption.

“Contributions of each author to the manuscript. Author contributions to individual tasks are marked with x.”

- References: Style of the list should be corrected to the journal style (e.g. journals names in abbreviated format)

28. We changed the reference style to match the Journal’s request.

- References: 19 - a typo (1?) in the publication title

29. We corrected this typo.

“19. Staude, I. R. *et al.* Disentangling non-random structure from random placement when estimating β -diversity through space or time. *Nat Ecol Evol* **4**, 802–808 (2020).”

- References: 23 - year of the publication should be placed in the end

30. We corrected this.

“23. Tilman, D. *Plant strategies and the dynamics and structure of plant communities.* Princeton University Press. (1988)”